

# Online gas and particle phase measurements of organosulfates, organosulfonates and nitrooxyorganosulfates in Beijing utilizing a FIGAERO ToF-CIMS

Michael Le Breton[1], Yujue Wang[2], Åsa M Hallquist[3], Ravi Kant Pathak[1], Jing Zheng[2], Yudong Yang[2], Dongjie Shang[2], Marianne Glasius[4], Thomas J Bannan[5], Qianyun Liu[6], Chak. K. Chan[7], Carl. J. Percival[8], Wenfei Zhu[9,] Shengrong Lou[9], David Topping[5], Yuchen Wang[6], Jianzhen Yu[6], Keding Lu[2], Song Guo[2], Min Hu[2] and Mattias Hallquist[1]

[1] Department of Chemistry and Molecular Biology, University of Gothenburg, Gothenburg, Sweden
[2] State Key Joint Laboratory of Environmental Simulation and Pollution Control, College of Environmental Sciences and Engineering, Peking University, Beijing, China
[3] IVL Swedish Environmental Research Institute, Gothenburg, Sweden
[4] Department of Chemistry and iNANO, Aarhus University, 8000 Aarhus C, Denmark
[5] Centre for Atmospheric Science, School of Earth, Atmospheric and Environmental Science, University of Manchester, Manchester, UK
[6] Division of Environment and Sustainability, The Hong Kong University of Science and Technology, Clearwater Bay, Kowloon, Hong Kong
[7] (School of Energy and Environment, City University of Hong Kong, Hong Kong
[8] Jet Propulsion laboratory, Pasadena, California, USA.
[9] Shanghai Academy of Environmental Sciences, Shanghai 200233, China

*Correspondence to:* M. Le Breton (Michael.le.breton@gu.se)

**Abstract**. A Time of Flight Chemical Ionisation Mass spectrometer (CIMS) utilizing the Fast Inlet for Gas and AEROsol (FIGAERO) was deployed at a regional site 40 km north west of Beijing and successfully identified and measured 17 sulfur containing organics (SCOs = organo/nitrooxyorgano sulfates and sulfonates) with biogenic and anthropogenic precursors. The SCOs were quantified using laboratory synthesized standards of lactic acid sulfate and nitrophenol organosulfate (NP OS). The mean total (of the 17 identified by CIMS) SCO particle mass concentration was $207 \pm 106$ ng m$^{-3}$ and had a maxima of 542 ng m$^{-3}$, although contributed to only $2 \pm 2.8\%$ of the organic aerosol (OA). SCO contribution to submicron mass (PM$_1$) indicates a dominant secondary production of SCO due to the low contribution of SCOs to PM$_1$ during periods of high mass loading. The CIMS identified a persistent gas phase presence of SCOs in the ambient air, which was further supported by post campaign vapour pressure measurements of NP OS. An increase in RH appeared to promote partitioning to the particle phase whereas higher temperatures favored higher gas phase concentrations. On average 12% of the total SCOs were observed in the gas phase with $C_{10}H_{16}NSO_7$ having just 5% and IEPOX OS having 44% on average in the gas phase.

Biogenic emissions contributed to only 19% of total SCOs detected. $C_{10}H_{16}NSO_7$, an alpha-pinene derived SCO, representing the highest fraction (10%) followed by an isoprene-derived SCO. Anthropogenic SCOs with PAH and





aromatic precursors dominated the SCO mass loading (51% total SCOs) with $C_{11}H_{11}SO_7$, derived from methyl napthalene oxidation, contributing to 40 ng m$^{-3}$ and 0.3% of the OA mass. Biomass burning was also identified as a potential anthropogenic and biogenic source of SCOs, based on nitrophenol (NP) and acetonitrile time series via secondary production of NP OS.

Gas and particle phase measurements of glycolic acid suggest that partitioning towards the particle phase promotes glycolic acid sulfate production, contrary to the current formation mechanism suggested in the literature. Highly oxidised multifunctional organic compounds (HOMS) and $RO_2$ radical diurnal profiles, as measured by the iodide ToF-CIMS, are similar to that of total SCOs, supporting results that indicate HOMS are able to play a role in SCO production. Anthropogenic related SCOs correlated well with benzene, although their abundance depended highly on the photochemical age of the air mass, tracked using the ratio between pinonic acid and its oxidation product, acting as a qualitative photochemical clock. The $HSO_4.H_2SO_4^-$ cluster measured by the CIMS was utilized as a qualitative marker for acidity and provides further evidence that the production rate of total SCOs is efficient in highly acidic aerosols with high $SO_4^{2-}$ and organic content. This dependency becomes more complex when observing individual SCOs due to variability of specific VOC precursors.

## 1. Introduction

Atmospheric particulate matter (PM) is well known to play a major role in adversely affecting the air quality and climate leading to severe health issues, such as respiratory and cardiovascular degradation (Pope *et al.*, 2002; 2011; Kim *et al.*, 2015). Secondary organic aerosols (SOA), formed through reactions of volatile organic compounds (VOCs) yielding semi volatile products that partition into the aerosol phase, represents a significant fraction of PM (Hallquist *et al.*, 2009) and remains the most poorly understood PM source (Foley *et al.*, 2010) due to the complexity of its chemical nature, resulting in discrepancies between observations and models (Heald *et al.*, 2005). Annual average $PM_1$ (particulate matter of diameter less than 1 micron) concentrations in Beijing reached 89.5 μg m$^{-3}$ in 2013 and, although recently dropped to 80.6 μg m$^3$, is still significantly above the Chinese National Ambient air quality Standard (CNAAQS, 35 μg m$^3$ annual average). The knowledge gap of emissions and PM primary and secondary production limits scientifically based abatement strategies targeting effects of secondary pollution that significantly contributes to PM in highly polluted regions (Hallquist *et al.*, 2016; Zhang *et al.*, 2012a). Therefore, Beijing is an ideal case study region for intense measurement campaigns to increase our understanding of the sources and processes involved in atmospheric aerosol chemistry in megacities and therefore providing the information necessary to implement mitigation strategies for Pan-Euroasian and Chinese societies. A growing number of field studies in this region have been performed in recent years, specifically focused on the haze events investigating the composition of primary and secondary particle aerosols and their formation mechanisms (Guo *et al.,* 2012, 2014, 2013; Huang *et al.*, 2010; Hu *et al.,* 2016, 2017; Li *et al.,* 2017).



Organosulfates (OSs) are known important SOA components formed by reactions between reactive organic compounds and sulfate (Iinuma *et al.,* 2007; Surratt *et al*., 2007, 2008), which is generated by the oxidation of $SO_2$, primarily emitted by fossil fuel combustion (Wuebbles and Jain, 2001). OSs have previously been measured in ambient aerosols at a number of varying geographical locations, from the remote to highly populated urban environments (Surratt *et al*.,

2007, 2008; Kristensen *et al.,* 2011; Stone *et al*., 2012; Zhang *et al*., 2012b; Worton *et al*., 2013; Shalamzari *et al*., 2013; Hansen *et al*., 2014) and also estimated using an AMS (Huang *et al*., 2015), although their composition and contribution to organic mass can vary significantly. To date, a large number of OSs have been identified across the world, although not all their precursors are known (e.g. Hansen *et al*., 2014). Mechanistic studies reveal multiple possible pathways for OS formation, which depend on availability of reactants in the atmosphere (Hettiyadura *et al*.,

2015), increasing the complexity of understanding their occurrence and abundance within models. Measurements of individual OSs have shown they may individually contribute up to 1% of the total organics (Olson *et al*., 2011; Liao *et al.,* 2015), indicating that the total OS contribution is comprised of many different species, which in turn will differ geographically due to emission sources and reaction pathways.

Isoprene OSs are hypothesized to be the most abundant in the ambient atmosphere (Surratt *et al*., 2007; Liao *et al.,*

2015) and are often used as markers of isoprene-derived SOA in field campaigns (Zhang *et al*., 2012b). Aromatic OSs have been recently observed in Lahore, Pakistan (Stone *et al*., 2012) and in urban sites in East Asia (Lin *et al*., 2012) and are considered to originate from anthropogenic sources. Riva *et al*. (2015 and 2016) have also previously probed the OS potential from PAH and alkane oxidation in the prescence of acidic sulfate aerosols. Glycolic acid sulfate (GAS) is considered another potentially important OS due to its common abundance and possible sources (Olson *et al*., 2011;

Liao *et al*., 2015). It is thought to form via a gas phase precursor reaction with an acidic aerosol sulfate and also from the particle phase reaction of methyl vinyl ketone with a sulfate particle, although both of these mechanisms are yet to be proven (Liao *et al*., 2015). GAS is also the only OS to date, which has been detected in the gas phase (Ehn *et al*., 2010), providing possible importance of gas to particle phase partitioning of some OSs.

OSs are thought to be good tracers for heterogeneous aerosol phase chemistry and SOA formation since the known

formation mechanisms involve reactive uptake of gas phase organic species onto aerosol (Surratt *et al.,* 2010). Due to their hydrophilic nature, polarity and relatively low volatility, they may significantly help nanoparticle growth and increase their potential to become cloud condensation nuclei (Smith *et al*., 2008). Therefore, it is imperative to improve our knowledge of OS abundance, formation, distribution, precursors and fate to help develop our understanding of SOA formation.

Mass spectrometry coupled with electrospray ionization is a common method to detect OSs (Iinuma *et al*., 2007; Reemtsma *et al*., 2006; Surratt *et al*., 2007; Gomez-Gonzalez *et al*., 2008) via the identification of the bisulfate ion, a fragment ion of OSs. Liquid chromatography mass spectrometry is known to efficiently separate aromatic and monoterpene derived organic sulfates containing aromatic rings or long alkyl chains and is therefore another suitable technique for OS detection (Stone *et al*., 2012). Hydrophilic interaction liquid chromatography has also been utilized



as a selective technique due to its ability to allow the OS to retain a carboxyl group, enabling detection of a larger suite of compounds (Gao *et al.,* 2006). The methods above often rely on sampling filters taken in the field and therefore provide a relatively low measurement frequency. This can limit the ability to evaluate production pathways when concentrations are often integrated over a period of hours. Further reactions on filters between the organics and sulfates

has also been postulated to add a bias to the OS concentration measured with respect to initial deposition onto the filter (Hettiyadura *et al*., 2017; Kristensen *et al*., 2016). Recently, a Particle Analysis Laser Mass Spectrometer (PALMS) was utilized to measure a number of OSs over the United States highlighting the ability of time of flight mass spectrometers to measure several OSs at high time frequencies (Liao *et al*., 2015).

This study utilizes a Filter Inlet for Gas and AEROsol (FIGAERO) Time of Flight Chemical Ionisation Mass

Spectrometer (ToF-CIMS) for the measurement of ambient sulfur containing organics (SCOs) at a semi-rural site 40 km from Beijing, China. This instrument enables sampling of either the gas-phase components or thermally desorbed particles by a high resolution mass spectrometer via a multi-port inlet, as described in detail by Lopez-Hilfiker *et al.,* (2014). The soft and selective ionization technique and high time resolution coupled with the FIGAERO enables the simultaneous detection and measurement of SCOs in the gas and particle phase to ng m$^{-3}$ concentrations. This work

aims to identify dominant SCOs in Beijing and their precursors. The high time resolution measurements are utilized to further probe their formation and abundance under different chemical and environmental regimes.

## 2. Experimental

### 2.1 Site description

The data presented here was collected during the measurement campaign "Photochemical smog in China" with an initiative to enhance our understanding of SOA formation via photochemical smog in China (Hallquist *et al.*, 2016). The campaign was coordinated by Peking University and University of Gothenburg with focus on spring/summertime episodic pollution episodes in Northeast China through gas and particle phase measurements. Here we briefly describe the measurement site and instrumental setup for data which is incorporated in this work. The setup was situated at a

semi-rural site 40 km North East of downtown Beijing close to Changping town (40.2207° N, 116.2312° E). All on-line instruments sampled from inlets on the 4$^{th}$ floor laboratory (12 metres above ground) at Peking University Changping Campus from the 13$^{th}$ May to 23$^{rd}$ June 2016, while filter measurements took place on the roof. The average temperature and humidity throughout the campaign were 23°C and 44% respectively. The wind speed averaged at 2 ms$^{-1}$ from the South-South West. A total of 4 pollution episodes were observed during the campaign period, which are

classified as sustained periods of high aerosol loading reaching a maximum of 115 micrograms per cubic meter (μg m$^{-3}$). The episodes were dominated by organic and nitrate aerosols although episode 3 contained high sulfate loading, equal to that of nitrate. The mass loading for the semi-rural site showed good correlation with the city campus



measurement site (30 km South-South-West of the Changping site and 12 km North West of downtown Beijing) throughout the campaign allowing for extrapolation of the semi-rural site results to inner city conditions. HYSPLIT back trajectory results showed the pollution episodes often correlated with air masses coming from the direction of Beijing (South-South-East). Clean air days were mostly North Westerly winds with clean air coming from the rural

mountain regions North West of Beijing and the measurement site.

A high resolution (4000) Time of Flight Aerosol Mass Spectrometer (ToF-AMS) was utilized to measure the mass concentrations and size distributions of non-refractory species in submicron aerosols, including organics, sulfate, ammonium and chloride (DeCarlo *et al., 2006*; Hu *et al.,* 2013). The setup of this instrument has been previously described by Hu *et al.* (2016). An Ionicon Analytik high sensitivity PTR-MS (Proton TRansfer Mass Spectrometer) as

described by de Gouw and Warneke *et al.,* (2007) provided supporting precursor VOC measurements.

**2.3 ToF-CIMS setup**

Gas and particle phase ambient species were measured using an iodide ToF-CIMS coupled to the FIGAERO inlet (Lopez-Hilfiker *et al*., 2014). The ToF-CIMS can be operated in either negative or positive ionization modes, and a

variety of reagent ion sources can be used. In this work the ToF-CIMS was operated in single reflection mode. The negative Iodide ion ($I^-$) was used as the reagent in all experiments. Dry UHP $N_2$ was passed over a permeation tube containing liquid $CH_3I$ (Alfa Aesar, 99%), and the flow was passing a Tofwerk X-Ray Ion Source type P (operated at 9.5 kV and 150 µA) to produce the ionization ions. The ionized gas was then carried out of the ion source and into the IMR through an orifice (Ø = 1 µm). Reaction products (e.g., compound X) were identified by their corresponding

cluster ions, $XI^-$ or the deprotonated ion, allowing for the collection of whole-molecule data. The nominal reagent and sample flow rates into the Ion-Molecule Reaction (IMR) chamber of the instrument were 3.5 liters per minute (LPM) and 2 LPM respectively. The IMR itself was temperature controlled at 40°C and operated at a nominal pressure of 500 mbar.

**2.4 FIGAERO inlet**

The FIGAERO inlet collected particles on a Zefluor® PTFE membrane filter. The aerosol sample line was composed of 12 mm copper tubing, while 12 mm Teflon tubing was used for the gas sample line. The FIGAERO was operated in a cyclic pattern; 25 minute of gas phase sampling and simultaneous particle collection, followed by a 20 minute period during which the filter was shifted into positioned over the IMR inlet and the collected particle mass was desorbed.

Desorption was facilitated by a 2 LPM flow of heated UHP $N_2$ over the filter. The temperature of the $N_2$ was increased from 20 to 250°C in 15 minutes (3.5°C min$^{-1}$), followed by a 5 minute temperature soak time to ensure that all remaining mass that volatilizes at 250°C was removed from the filter. The resulting desorption time series profiles allowed for a




distinct separation of measured species as a function of their thermal properties. The ToF-CIMS was configured to measure singularly charged ions with a mass to charge ratio (*m/z*) of 7 – 620, a reduced mass range in order to compensate for the lower count rate emitted by the soft X-ray source with respect to the Polonium-235 radioactive source as commonly deployed. The mass range was changed during the campaign to view to a higher mass range (1000 *m/z*) to ensure no major contributing peaks were being unaccounted for. Perfluoropentanoic acid was utilised as a mass calibrant up to *m/z* 527 through its dimer and trimer. This range of mass calibration peaks also limited accurate peak identification above *m/z* 620.

### 2.5 Knudsen Effusion Mass Spectrometer (KEMS)

The KEMS technique was utilized to measure the vapour pressure (vapour pressure) of potential gas phase SCOs measured by the CIMS. This technique is able to measure vapour pressures from $10^{-1}$ to $10^{-8}$ Pascals (Pa) ranging from volatile organic compounds to extremely low volatility organic compounds. A full description of the technique can be found in Booth *et al*. (2009, 2010) and the measurements of a series of compounds over a large VP range, in a recent inter-comparison study from this instrument can be found in Krieger *et al*. (2017). Briefly, the instrument consists of a temperature controlled Knudsen effusion cell, suitable for controlled generation of a molecular beam of the sample organic compounds in a vacuum chamber, coupled to a quadrupole mass spectrometer. The cell has a chamfered effusing orifice with a size ≤1/10 the mean free path of the gas molecules in the cell. This ensures the orifice does not significantly disturb the thermodynamic equilibrium of the samples in the cell (Hilpert, 2001). The system is calibrated using the mass spectrometer signal from a sample of known vapour pressure, in this case malonic acid (vapour pressure at 298K = $5.25 \times 10^{-4}$ Pa, (Booth *et al*., 2012)). A load-lock allows the ioniser filament to be left on, then a new sample of unknown vapour pressure can be measured. Solid state vapour pressures measured in the KEMS can then be converted to sub-cooled liquid vapour pressures using the melting point, enthalpy and entropy of fusion, which are obtained by using a Differential Scanning Calorimeter (DSC) (TA instruments Q200).

### 2.6 SCO identification

Peak fitting was performed utilizing the Tofware peak fitting software for molecular weights up 620 AMU. Unknown peaks were added to each peak on the spectra until the residual was less than 5%. Each unknown peak was assigned a chemical formula using the peaks exact mass maxima to 5 decimal places and also isotopic ratios of subsequent minor peaks. An accurate fitting was characterized by a ppm error of less than 5 and subsequent accurate fitting of isotopic peaks. An example of the spectra and peak fitting can be found in Figure 1, highlighting the mass spectral fit for GAS and nitrophenol sulfate (NP OS), previously identified ambient SCOs. Although the structure cannot be determined with CIMS, it is assumed that no fragmentation of larger SCO species contribute to the SCO identified due to the soft



ionization technique employed. The SCOs were identified in the spectra as negative ions assumed to be formed by hydrogen removal. Here, we present 17 SCOs that were identified in the mass spectra, which are displayed in Table 1 with their respective exact mass, formula, literature nomenclature and possible precursors. The SCOs detected ions ranged from 154.96 *m/z* (GAS) to 294.06 *m/z* ($C_{10}H_{11}NSO_7$). The number of oxygen in the SCO ranged from $O_3$ ($C_7H_7SO_3$) to $O_7$ ($C_5H_8SO_7$). Filter measurements were taken diurnally at the same sampling site, although from a different inlet and location in the building, to which orbitrap and HPLC MS analysis was performed to identify OSs present in the ambient air. The CIMS measurements generally agreed well, once processed into diurnal loadings, which confirms the correct identification and measurement of SCOs by CIMS. This analysis is not within the scope of this work and provides the basis of the correct identification to which a future paper will probe the caveats observed between the measurement techniques.

It is acknowledged that the CIMS may not detect all SCOs in the ambient air due to peak fitting resolution limitations and limits of detection, therefore enabling the possibility for misrepresentation of the dominant SCO and an underestimation of total abundance, although for the analysis in this work it is assumed that the measured SCOs do represent a significant fraction. No physical features of the SCO (structure, O:C ratio, mass etc.) should inhibit the CIMS identifying the major SCO in the Beijing ambient air.

## 2.7 Quantification of SCOs

The OS and NOS calibrations were performed after the campaign and normalized to the in-situ formic acid calibrations (as described in le Breton *et al*., 2012, 2013) to account for any drift in sensitivity throughout the campaign. This relative sensitivity technique has been previously utilized for $N_2O_5$ and $ClNO_2$ and has been verified with laboratory experiments (Le Breton *et al*., 2014). As a result of low mass range of the SCOs, common functionality, relatively small change in polarity and lack of available stable SCO standards, we calibrated for 2 SCOs (lactic acid sulfate (LAS) and NP OS) and applied an average sensitivity for all the SCOs detected in Beijing. The ToF-CIMS sensitivity utilizing iodide as a reagent ion is known to vary by up to 3 orders of magnitude; therefore, further work is necessary to develop SCO standards and assess possible variations in sensitivity. NP OS is available commercially from Sigma Aldrich and was utilised to calibrate for the NOS's. L(+) Lactic acid from Sigma Aldrich (95%) was utilised as the preliminary agent for lactic acid sulfate synthesis and was produced using the same technique as Olson *et al*. (2011). Briefly, a solution of 76.1 mg, 1.29 mmol, lactic acid in 2 mL di-methyl-formamide (DMF) was added dropwise to trioxide pyridine (0.96g, 7.75mmol) in 2mL DMF at 0 °C. The solution is then stirred for 1 hour at 0 °C and 40 minutes at room temperature, the solution is re-cooled to 0 °C and trimethylamine (0.23 mL, 1.66 mmol) was added for quenching and the mixture wad further stirred for 1 hour. The solvent is then evaporated under vacuum and NMR is directly utilised to calculate the purity which was found to be 8.2%.





A known mass of the solid calibrant (NP OS and Lactic acid sulfate) was added to 3 different volumes of milliQ water to produce different concentration standards. A known volume of each solution was then placed onto the FIGAERO filter and a desorption cycle was performed. The total ion counts for the HR SCO peak relates directly to the sensitivity of the system with respect to total ion counts per molecule reaching the detector. Figure 2 shows a 3-point calibration curve for NP OS and the corresponding thermogram, mass spectra and peak fit. The sensitivity of LAS and NP OS calibrations were calculated to be 2 and 1.6 ion counts per ppt Hz$^{-1}$ respectively. All SCO were calibrated using the LAS sensitivity and all NOS using the NP OS sensitivity.

During desorption of both SCOs, fragmentation of the organic core and sulphate group was observed resulting in a desorption profile of at *m/z* 97 (the bisulphate ion) and the deprotonated organic mass, i.e. $C_3H_5O_3$ for lactic acid. A number of different temperature ramping rates were performed with the FIGAERO to further probe the fragmentation and highlighted an increase in ramp rate (°C/minute) decreased the calculated sensitivity due to an increase in fragmentation. This not only serves to highlight how the calibration tests of a species must mimic the exact measurement conditions, but also suggests potential interferences from fragmentation on the organic *m/z*´s. The relatively low concentration of the OS with respect to the organic precursor results in little error in quantification, although this ratio may significantly change in different air masses and a number of products of organic oxidation may fragment resulting in a significant error. This fragmentation can also be observed within the high resolution thermograms of the FIGAERO as a double desorption and further highlights the necessity for detailed thermogram analysis to accurately deconvolve desorption's relevant only to particle loss from the filter and not fragmentation or ion chemistry in the IMR. The fragmentation is considered to be constant throughout the campaign. The error for the SCO measurements will vary for each individual OS. Here we calculate an average error of 52 % for the SCOs, calculated using the standard deviation of the NP OS calibration time series data.

The limitation of FIGAERO temperature ramps to 250 °C may result in further error as some SCOs may not be fully desorbed from the filter due to their low vapour pressures. To evaluate the mass left on a filter, several double desorption cycles were performed where mass is collected and desorbed such as in standard use. This is performed by re-heating the same filter once cooled to attain a second thermogram of the same filter. The second thermogram exhibited an average of 90% reduction of counts for the OS, although the NOSs had an average decrease of 82% counts. This indicates that most, but not all mass, are removed from the filters when desorbing. For the interpretation of the results of the field campaign this effect will induce a small distortion on the time evolution of SCOs when comparing to other parameters, e.g. 9% of NOSs will remain on the filter and being subjected to the subsequent desorption cycle.



## 3. Concentrations and partitioning of atmospheric SCOs

### 3.1 SCO contribution to PM$_1$ at Changping

The SCOs measured at the Changping site had a mean campaign concentration of 207$\pm$ 106 ng m$^{-3}$ (Table 2). The highest concentration of total SCOs during the campaign was 542 ng m$^{-3}$ and the lowest 40 ng m$^{-3}$, thus they are
omnipresent and have significant sources during most atmospheric conditions. These concentrations are consistent with Stone *et al*. (2012) reporting an average OS concentration of 700 ng m$^{-3}$ in a number of rural and urban sites in Asia. A mean SCO contribution to OA in the work presented here was calculated to be 2 $\pm$ 2.8% (Table 2), within the range of values calculated by Stone *et al.* (2012) (0.8% to 4.5%), further supporting evidence that SCO contribution to PM$_1$ mass is relatively low in Asia. The CIMS cannot claim to measure total SCO, rather than singularly identify and measure
SCOs contributing to the total mass loading. Therefore, the SCO contribution reported in this work should be considered as a lower limit. The Liao *et al*. (2015) study also supports the idea that SCO contribution to PM$_1$ mass in anthropogenically dominated regions is less significant than that from biogenically dominating air masses by observing a significantly higher contribution of IEPOX sulfate to PM$_1$ mass on the East coast of the United States (1.4%).
The observation of higher relative contribution of SCOs to total organics in more remote regions compared to a densely
populated urban area, supports the idea that SCOs provide a higher contribution to mass in aged air due to their secondary production pathways. Similar to Lahore (as studied by Stone *et al*., 2012), Beijing has many strong primary anthropogenic sources which will dominate the mass loading and therefore, initially, will contribute to a lower fraction of the total concentration from SOA due to limited processing near the source. Throughout the campaign, a good correlation (R$^2$ = 0.66) was observed between an increase in ΔSCO mass and PM$_1$ mass, although the OS contribution
to PM$_1$ decreased exponentially (Figure 3) indicating that the pollution episodes contain a lower fraction of SCOs with respect to total PM$_1$ than the clean air days and that high pollution containing air masses contained more OSs but at a less significant fraction. This reuslt also suggests trhat SCO do not play as large a role as expected even though their precursors (organics and sulfate) ar abundant within the episodes, indicating the conditions of their formation may be more vital than absolute concentrations of precursors.

### 3.2 Gas to particle phase partitioning of SCOs

The FIGAERO ToF-CIMS data exhibited indication of SCOs in both the particle and gas phase. Once all HR peaks have been identified, the batch fitting and HR time series for the whole data set is processed and then separated into gas phase measurements and particle phase desorption profile time series. Subtraction of both the gas phase background
periods and blank filter desorption's removes any data underneath the detection threshold of ambient gas species and particles. Upon analysis of the resultant data, significant concentrations of gas phase SCOs were observed. Figure 4 depicts the overall sum SCO mass concentration time series in the gas and particle phase. The mean contribution from





gas phase SCO to total SCO was found to be 11.6%, 23±8 ng m$^{-3}$. This suggests a significant amount of OS is always present in the gas phase and factors that influence gas-to-particle partitioning sustain this significant contribution. The R$^2$ correlation between gas phase and particle phase SCOs is 0.43 that suggests reliability upon total SCO for partitioning but indicates a more complex system that governs the partitioning. This low correlation also further

supports the probability that the concentrations do not results from an instrument artefact as this would likely to result in a constant ratio of particle to gas phase concentrations. Previous studies have supported the existence of gas phase GAS in ambient air (e.g. Ehn *et al*., 2010), although some work has attributed other measurement techniques detection of gas phase SCO to result from measurement artefacts (Hettiyadura *et al*., 2017; Kristensen *et al*., 2016). One possibility is the deposition of SCOs onto the IMR walls during the temperature ramp of the desorption which in time

may de-gas and be observed in the gas phase. This would likely to cause a hysteresis in the observed gas phase measurements with respect to the particle phase, which was not observed.

The vapour pressure of NP OS was measured using the KEMS instrument in the laboratory to establish the possible existence of gas phase SCOs. This technique has recently been employed to measure the vapour pressure of NP (Bannan *et al*., 2017). The KEMS experiments found the solid state vapour pressure of NP OS to be 5.07×10$^{-5}$ Pa at 298 K.

Assuming an average subcooled liquid correction for all compounds measured in the Bannan *et al*. (2017) study, as no DSC data is available, the subcooled liquid vapour pressure of NP OS is 2.32x10$^{-4}$ (Pa). This vapour pressure lies within the semi volatile organic compound range, therefore supporting the potential partitioning of OSs to the gas phase under ambient conditions. Analysis of each OS T$_{max}$ from the FIGAERO indicates that more than half of the SCOs detected have a higher vapour pressure than NP OS which also indicates a significant partitioning towards the gas phase. This

is an unexpected result due to the lack of literature indicating OSs have a gas phase presence. To further validate the CIMS and KEMS findings, one can further calculate the VPs from the FIGAERO data utilizing the T$_{max}$ and compare these to literature values, confirming that the partitioning calculated by CIMS is representative for compounds well established within the literature. The CIMS calculated VPs of malonic, succinic and glutaric acid were 2x10$^{-3}$, 1.85x10$^{-3}$, 1x10$^{-3}$ Pa which compare well to the Bilde *et al*. (2015) VPs; 6.2x10$^{-3}$, 1.3x10$^{-3}$, 1x10$^{-3}$. This agreement further

supports the NP OS KEMS VP and therefore the partitioning observed by CIMS for uncalibrated OSs. Aerosol liquid water content will affect the partitioning of gas phase compounds to aerosols (Zhang *et al*., 2007). Data point size coding the correlation of the gas and particle phase SCO concentrations indicates partitioning towards the aerosol phase at lower relative humidities (Figure 4). Conversely, as temperature increases (as indicated by red colour shading) the SCOs partition further towards the gas phase, as thermodynamically expected.

NOS C$_{10}$H$_{16}$NO$_7$S displayed the highest particle to gas phase ratio of 13.9 and has mean presence in the gas phase of 4.4%. The most prominent gas phase SCO particle to gas phase ratio was IEPOX OS which has a particle to gas phase mean ratio of 1, indicating, on average, this SCO is evenly distributed between the gas and particle phase. GAS, as measured in the gas phase in previous literature (Ehn *et al*., 2010) has a mean particle to gas phase ratio of 10.3, contributing to 7.1% of GAS from the gas phase with a maximum campaign contribution in the gas phase of 29.5%.



Further work is necessary to validate these findings and determine the mechanisms and importance of gas phase SCO abundance in ambient air. The significant contribution of IEPOX-sulfate in the gas phase is an unexpected result, which could have serious implications on our knowledge of SOA formation. One possible, explanation is a gas phase formation pathway of IEPOX sulfate. The high contribution in the gas phase could also be further perturbed if an equilibrium between condensation to particle phase and gas phase formation has not been established. It must be noted that the VP of these compounds has not been calculated and correct calibration of $T_{max}$ is necessary to extract such information, but qualitatively the relative VP compared to NP OS can be utilized as a reliable scale due its independent calibration by KEMS.

## 4. Sources and secondary formation of SCOs

### 4.1 SCO sources at the Changping site

SCOs are known to have biogenic and anthropogenic sources and some which have multiple sources from both, e.g. GAS (Hettiyadura *et al.,* 2017; Hansen *et al.*, 2014). Burning events are known to emit high levels of organics and nitrates and potentially sulfur, depending on the type of fuel used. This enables biomass burning to be a potential anthropogenic and biogenic source of SCOs through release of their precursors. The site at Changping was influenced by both regional anthropogenic pollution from the Beijing area and localized anthropogenic activity (industry, biomass burning and traffic) but also emissions from biogenic sources, as it is situated in a semi-rural area, with forest, vegetation and plantations. This was evident from the toluene and isoprene proton transfer mass spectrometer (PTR-MS) measurements which have mean campaign concentrations of $0.55 \pm 0.4$ and $0.27 \pm 0.19$ ppb respectively with maxima of 5 and 1.5 ppb respectively. Thus, as shown in Figure 5, PTR-MS measurements of benzene and isoprene were utilized to evaluate and exemplify the attribution of aromatic and biogenic mass contribution of SCOs measured in this work. Data on days with incomplete time series have been removed to ensure the data presented represents a full mean of the day concentration. A good correlation between the benzene:isoprene ratio and sum of SCOs is observed suggests an increase in relative anthropogenic emissions promotes an increase in total SCO loading. It should be noted that $C_6H_{10}SO_7$ has no known precursor in the literature, although it contributes significantly to the SCO mass loading in this work (16%).

### 4.1.1 *Biogenic and anthropogenic SCOs*

Biogenic SCOs are known to be comprised of monoterpene and isoprene derived SCOs which have been identified in rural, sub-urban and urban areas around the world, and have been shown to be a major constituent of SOA (Surratt *et al*., 2008; Liao *et al., 2015*). IEPOX sulfate is commonly found to be the most dominant OS at many locations and was identified in the spectra at the Changping site. The IEPOX sulfate mean concentration represented 0.11% of the OA



mass, agreeing well with concentrations found in Western USA (significant anthropogenic emissions) and lower than the Eastern USA as expected due to higher biogenic and isoprene emissions (Liao *et al.,* 2015). Although IEPOX sulfate is considered one of the most abundant individual organic molecules in aerosols (Chan *et al*., 2010), here our results show it only contributed to 2% of the SCO mass and was the 8[th] most abundant SCO in the particle phase. $C_5H_8SO_7$

and $C_4H_8SO_7$ were also two isoprene derived OSs measured by the CIMS with mean campaign concentrations of 2 and 3 ng m$^{-3}$ respectively and a contribution of 0.02% to OA mass. The highest contributing biogenic SCO to the ambient air was a NOS, $C_{10}H_{16}NSO_7$, a known NOS derived from alpha-pinene oxidation. This NOS had a mean campaign concentration of 21 ng m$^{-3}$ and a 0.2% contribution to OA mass.

Anthropogenic SCOs, including polyaromatic hydrocarbon (PAH) derived SCOs have received more attention in

research studies due to their identification and therefore can also contribute to altering SOA hygroscopicity and radiative properties via acting as surface reactants (Nozière *et al*., 2010; Hansen *et al*., 2015). Aromatic SCOs and sulfonates have only recently been identified as atmospherically abundant SCOs (Riva *et al.,* 2015). In this work we find that the PAH derived SCO $C_{11}H_{11}SO_7$ is the most dominant SCO in Beijing with a mean concentration of 40 ng m$^{-3}$, contributing to 20% of the total SCO mass and 0.4% of the OA mass. This SCO has been identified in laboratory studies as an SCO

forming from the photo-oxidation of 2-methyl napthalene, one of the most abundant gas phase PAHs and is thought to represent a missing source of urban SOA (Riva *et al.,* 2015). This work presents the possible significance of PAH SCOs in Beijing and further evidence that photo-oxidation of PAHs represents a greater SOA potential than currently recognized. A further 8 anthropogenic aromatic derived SCOs were identified as common components of the PM$_1$ representing more than half of the total SCOs with $C_7H_5SO_4$ contributing to 24 ng m$^{-3}$ and 0.23% OA mass. The total

anthropogenic related SCOs had a mean mass of 122 ng m$^{-3}$ and contributed to 1.2% of the OA mass.

### 4.1.2 *Biomass burning source of SCOs*

NP (a product of benzene oxidation and nitration) has previously been detected in the gas and aerosol phase (Harrison *et al.,* 2005) and is an important component of brown carbon (Mohr *et al.,* 2013). NP has primary sources, such as

vehicle exhausts and biomass burning (Inomata *et al.,* 2013 and Mohr *et al.,* 2013) and secondary sources via the photo-oxidation of aromatic hydrocarbons in the atmosphere (Harrison *et al.,* 2005). High levels on anthropogenic activity, biomass burning and strong photochemistry in Beijing therefore enable this region to be a strong potential source of NP. Diurnal profiles of NP exhibit a significant increase in the morning, possibly due to both local biomass burning and vehicle emissions, which is maintained throughout the day. A second large increase in concentration is observed

around 4pm, likely due to vehicle activity. The NP OS also exhibits a similar diurnal structure, although its concentration builds up more rapidly in the afternoon and early evening, likely to be a result of the photochemical aging of NP and eventual reaction with SO$_4^{2-}$ to produce the OS. The campaign time series for NP and NP OS can be seen in Figure 6. Unlike its precursor and most other pollutant markers measured in this work, including all other SOCs, NP





OS exhibits higher concentrations between 28th to the 1st June compared to the 17th May to 21st May. The only compound with a similar campaign profile is acetonitrile (a marker for biomass burning), which has significantly enhanced concentrations between 6 and 8am from the 17th to the 21st May. Back trajectories of these two time periods show the air mass during the first period comes from the west, a more rural region of China and known to be influenced

heavily by biomass burning, whereas the second time period has wind directions mainly bringing in air masses that have gone through the Tianjin and Beijing area. It is therefore hypothesized that the NP OS, which peaks later in the day than the NP and acetonitrile, is a secondary product formed from the biomass burning events indicated by the acetonitrile data.

**4.2 SCO production mechanisms**

**4.2.1 *Precursor analysis***

The availability of the organic precursors of SCOs is a limiting factor for the SCO production rate. The measurement of the precursors in the gas and particle phase by CIMS enables a more descriptive mechanism to be outlined as the partitioning of the precursor will vary the distribution between gas and particle production pathways and therefore rate

of corresponding SCO production. Glycolic acid has on average 75% of its mass in the gas phase for the measurement period whereas GAS is dominantly in the particle phase (Figure 7). The GAS particle phase concentration is observed to increase as the $SO_4^{2-}$ mass loading increases and the GA gas and particle concentrations increase, although the partitioning of the GA towards the gas phase restricts the OS production. This can be seen in Figure 7 as for a given $SO_4^{2-}$ concentrations, the data with warm colors (red), representing high fraction of precursor GA in particle phase,

generally provides higher concentration of particulate GAS.

The main formation mechanism of GAS is thought to be via the reaction of GA in the gas phase with an acidic aerosol sulfate (Liao *et al*., 2016), contrary to what is observed here. Although an increase in GAS is observed to correlate with the GA, it appears that a partitioning towards the particle phase promotes GAS production. An $R^2$ correlation of 0.68 between GAp and GASp whereas an $R^2$ of 0.4 is observed between GAg and GASp.

The sum of benzene SCOs exhibits a good correlation to the gas phase benzene time series (Figure 8), although their abundance should also rely on the availability of sulfur in the particle phase and the age of the air mass, if it is assumed that they are formed via secondary reactions of primary pollutants. In order to assess how the SCO production rates may vary due to these factors, two distinct high benzene SCO events with similar benzene concentrations were scrutinized, i.e. the 29th May and the 1st June. The first period has lower $SO_4^{2-}$ concentrations, higher $H_2SO_4$ levels and

a higher total benzene SCO concentration. The exact age (or time for oxidation) of compounds in an air mass are without an extensive modelling study complicated to derive. However, as proxies to attain an approximation about oxidation state one may use some trace compounds. Monoterpene oxidation by the hydroxyl radical (OH) or $O_3$ results in the





formation of multifunctional organic acids such as pinonic acid which can then be further oxidised by OH to form 3-methyl-1,2,3-butane-tricarboxylic acid (MBTCA), both of which are measured by CIMS. Therefore, in an air mass containing monoterpene emissions, as known here through the identification of their products such as $C_{10}H_{16}NO_7S$, we can utilize the ratio of pinonic acid and MBTCA; as tracers of monoterpene SOA processing as detected in ambient

aerosols in Europe, USA and Amazon (Gao *et al*., 2006) as a relative photochemical clock. During the high benzene event on the 1st June, according to the pinonic acid: MBTCA ratio, the air mass is less oxidized relative to the air mass on the 29th May (Figure 8). This would allow less time for secondary production and explain the relatively lower concentration of SCOs, irrespective of higher $SO_4^{2-}$ concentrations and similar benzene concentrations.

To further elaborate The Gothenburg Potential Aerosol Mass (Go:PAM) chamber was tested and utilized to simulate

aging of the air mass during periods of the campaign. Here the ratio between pinonic acid and MBTCA was observed to increase by an average of 3 during aging within the Go:PAM which has been calculated to be the OH exposure equivalent of 2 days in the ambient atmosphere. As this ratio increased with aging, the SCO concentration also increased exponentially, further supporting the secondary production of SCO in photochemically aged air mass. Although limited data is available here for simultaneous Go:PAM and CIMS measurements, the results indicate the potential utilization

of the chamber to probe secondary production processes.

**4.2.2** *Aerosol acidity*

The molecular ion $H_3S_2O_7^-$ was identified in the mass spectra throughout the campaign, which has previously been detected by Liao *et al*. (2015) using a Particle Analysis Laser Mass Spectrometer to measure SCOs. They attribute this

mass to be a cluster of $HSO_4^-$ with sulfuric acid ($H_2SO_4$). Particles in the presence of $H_2SO_4$, and therefore high acidity, form this cluster whereas neutralized ions are likely to favour the unclustered $HSO_4^-$ form. Therefore, the ratio between the cluster and the bisulfate ion increases with increasing aerosol acidity (Murphy *et al*., 2007; Carn *et al*., 2011). Liao *et al*. (2015) validate the appropriateness of this cluster as a marker for aerosol acidity thorough comparisons to a thermodynamic model with gas and aerosol phase measurement inputs. Acidity was also calculated utilizing the gas

and particle phase $H_2SO_4$ and liquid $H^+$ ion concentration analysed using an offline technique, as described by Guo *et al.* (2010), from diurnal samples taken at the site. This method showed good agreement with the integrated diurnal counts of the $H_3S_2O_7^-$ ion and therefore. Therefore, we employ the $HSO_4^-.H_2SO_4^-$ cluster in this work as a qualitative scale for particle acidity utilizing similar assumptions. Figure 9 shows how total SCO mass concentration generally increased as total organic mass from the AMS increased. The correlation indicates that higher acidity (darker colours)

tends to promote formation of SCOs when in the presence of high levels of organics and $SO_4^{2-}$ (larger symbol sizes), supporting the growing consensus that aerosol acidity plays an important role in ambient SCO formation. This importance of acidity agrees well with both the acid-catalyzed epoxydiol ring opening formation mechanism (Surratt



*et al*., 2010) and the sulfate radical initiated SCO formation because efficient formation of sulfate radicals also requires acidity (Schindelka *et al*., 2013).

## 5. Conclusions

The FIGAERO ToF-CIMS was successfully utilized for the ambient detection of 17 SCOs in Beijing in the gas and aerosol phase with limits of detection in the ng m$^3$ range. Further calibrations and comparisons to total SCO measurements are required to evaluate its performance limitation with regards to sensitivity application and peak identification. The SCOs measured by CIMS contributed to 2% of the OA at the semi-rural site, highlighting the relatively low contribution of SCOs in Beijing, an anthropogenically dominated environment. This calculation from

CIMS may only be valid to infer each individual SCO contribution to total SOC mass as limitations in SCO identification and quantification limit the CIMS ability for total SCO measurements. Significance of their secondary production pathway prevailed, although still present in relatively fresh air masses. Contributions of SCO to total organics (2±2.8%), sulfate (15±19%) and PM (1±1.4%) indicate the concentrations observed emissions in Beijing result from highly processed ambient air masses.

Gas phase SCOs were identified for all the SCOs measured at the site, contributing to in average to 18% of the total SCO mass. The possibility of gas phase OSs in ambient air was supported by KEMS vapour pressure measurements of NP OS which suggests a vapour pressure in the semi volatile range. A number of SCOs possessed T$_{max}$ values similar to that of NP OS indicating they should also have semi volatile vapour pressures which is indicated by their gas phase presence observed in this work. The partitioning towards the gas phase was more efficient at high atmospheric

temperatures, while lower relative humidities promoted partitioning to the particle phase.

Biogenic SCOs contributed to a small fraction of the total OS mass at Changping and was dominated by an alpha pinene derived OS with 0.2% contribution to the OA mass. IEPOX sulfate was only the 8[th] most abundant SCO measured, contrary to common reports that it is one of the most abundant OS within the aerosol. Anthropogenic precursors contributed to more than half of the SCO mass loading with a PAH derived SCO contributing to as much as 1.2% of

the OA mass. Benzene derived SCOs correlated well with gas phase benzene levels and were heavily influenced by photochemical aging, as indicated by the pinonic acid: MBTCA ratio. The contribution of each benzene derived SCO to total benzene derived SCO mass varied daily and throughout the campaign highlighting the complexity of the atmospheric processing and composition of SC. Significant contributions from aromatic SCOs highlight the importance of anthropogenically emitted organics in the Beijing region and their contribution to the Beijing outflow and subsequent

photochemistry. NP OS was attributed to biomass burning emissions due to its campaign intensity coinciding with high levels of acetonitrile. This highlights the importance of anthropogenic emissions and their contribution to SOA from the urban Beijing outflow.



A qualitative CIMS marker for aerosol acidity highlighted the increase in SCO production rate in acidic aerosols in the presence of high $SO_4^{2-}$ and organics. The correlation of SCO production and RH becomes more complex for individual SCOs, which cannot be resolved within this studies framework.

**Acknowledgement:**

The work was done under the framework research program on 'Photochemical smog in China" financed by Swedish Research Council (639-2013-6917). The National Natural Science Foundation of China (21677002) and the National Key Research and Development Program of China (2016YFC0202003) also helped fund this work.

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



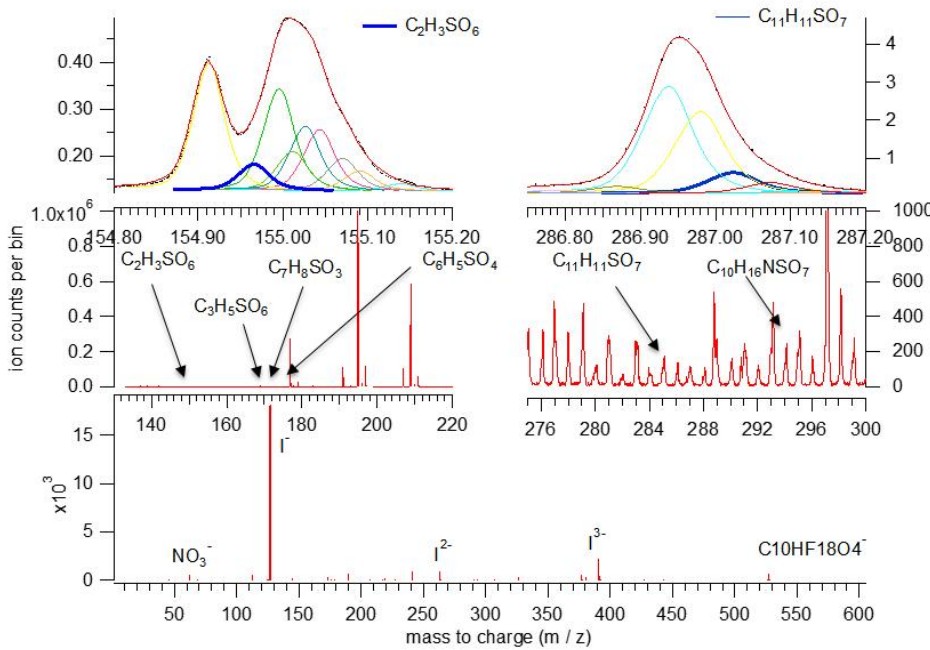

**Figure 1.** The bottom panel displays the average campaign mass spectra for the whole mass range of the ToF (3-620) which is further expanded to show small regions in the middle panel and specific and HR fitting for individual peaks in the top panel.



**Table 1. SCOs identified at the Changping site with their respective mass, chemical name and potential precursors.**

| m/z ion | Molecular formula | Reference | OS name | Precursor | [mean] µgm-3 | mean % PM | % OS |
|---|---|---|---|---|---|---|---|
| 217.9759 | $C_6H_4NSO_6^-$ | - | Nitrophenol sulphate | Nitrophenol | 2.60 | 0.00 | 0.4 |
| 154.965582 | $C_2H_3SO_6^-$ | Surrat 2007 | Glycolic acid sulphate | Glycolic acid | 9.60 | 0.03 | 1.6 |
| 168.981232 | $C_3H_5SO_6^-$ | Olson 2011 | Lactic acid sulphate | Lactic acid | 35.00 | 0.12 | 5.9 |
| 171.012139 | $C_7H_7SO_3^-$ | Riva 2015 | | Aromatics (Benzene and PAHs) | 16.00 | 0.06 | 2.7 |
| 172.019964 | $C_7H_8SO_3^-$ | Riva 2015 | | Aromatics (Benzene and PAHs) | 5.40 | 0.02 | 0.9 |
| 184.991403 | $C_7H_5SO_4^-$ | Riva 2015 | | Aromatics (Benzene and PAHs) | 72.00 | 0.25 | 12.1 |
| 187.007053 | $C_7H_7SO_4^-$ | Staudt 2014 | Methyl phenyl sulphate | benzene | 40.00 | 0.14 | 6.7 |
| 199.007053 | $C_8H_7SO_4^-$ | Riva 2015 | | Aromatics (Benzene and PAHs) | 14.00 | 0.05 | 2.4 |
| 199.999622 | $C_4H_8SO_7^-$ | Surrat 2007 | | 2-methylglyceric acid (isoprene) | 5.00 | 0.02 | 0.8 |
| 201.022703 | $C_8H_9SO_4^-$ | Staudt 2014 | 4 methyl benzyl sulphate | benzene | 22.00 | 0.08 | 3.7 |
| 211.999622 | $C_5H_8SO_7^-$ | Surrat 2008 | | isoprene | 8.00 | 0.03 | 1.3 |
| 215.023097 | $C_5H_{11}SO_7^-$ | Surrat 2010 | IEPOX sulphate | IEPOX | 27.00 | 0.09 | 4.5 |
| 226.015272 | $C_6H_{10}SO_7^-$ | Boris 2016 | unknown | unknown | 90.00 | 0.31 | 15.1 |
| 229.017618 | $C_9H_9SO_5^-$ | Riva 2015 | | Aromatics (Benzene and PAHs) | 30.00 | 0.10 | 5.0 |
| 231.033268 | $C_9H_{11}SO_5^-$ | Riva 2015 | | Aromatics (Benzene and PAHs) | 40.00 | 0.14 | 6.7 |
| 287.023097 | $C_{11}H_{11}SO_7^-$ | Riva 2015 | | Aromatics (Benzene and PAHs) | 120.00 | 0.41 | 20.2 |
| 294.065296 | $C_{10}H_{16}NSO_7^-$ | Surrat 2008 | | alpha pinene | 58.80 | 0.20 | 9.9 |



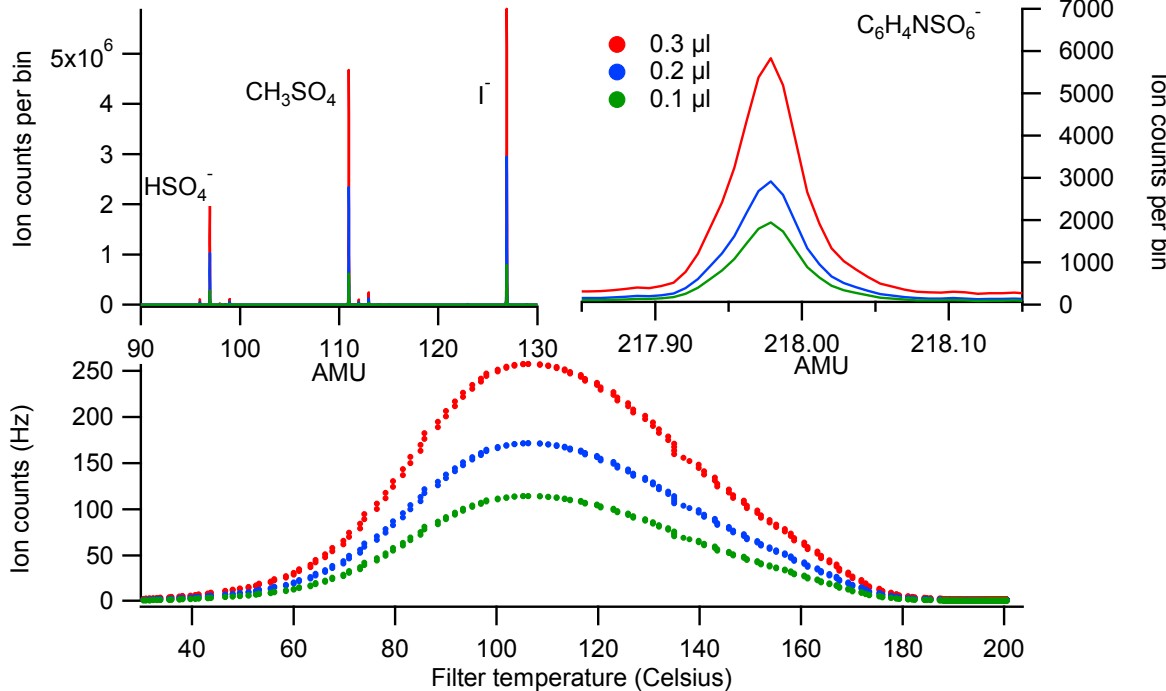

**Figure 2.** The time series of the NP OS 3 step calibration time series for 0.1 μl, 0.2 μl and 0.3 μl 1000 ppm solution is displayed in the bottom panel and its corresponding average stick (top left) and HR (top right) intensities and peak shapes for the desorption.

**Table 2.** The mean campaign mass, percentage contribution to $PM_1$ and percent contribution to total SCOs for 10 each SCO identified in this work.

| Molecular formula | [mean] ngm-3 | mean % OA | % SCO | Molecular formula | [mean] ngm-3 | mean % OA | % SCO |
|---|---|---|---|---|---|---|---|
| C11H11SO7- | 40 | 0.40 | 19.45 | C8H9SO4- | 6 | 0.06 | 3.05 |
| C6H10SO7- | 30 | 0.30 | 14.63 | C7H7SO3- | 6 | 0.06 | 2.89 |
| C7H5SO4- | 24 | 0.23 | 11.41 | C8H7SO4- | 5 | 0.05 | 2.25 |
| C10H16NSO7- | 21 | 0.20 | 9.97 | C7H8SO3- | 4 | 0.04 | 2.09 |
| C7H7SO4- | 14 | 0.14 | 6.75 | C2H3SO6- | 4 | 0.04 | 1.93 |
| C9H11SO5- | 13 | 0.13 | 6.27 | C4H8SO7- | 3 | 0.03 | 1.45 |
| C3H5SO6- | 13 | 0.12 | 6.11 | C5H8SO7- | 2 | 0.02 | 1.13 |
| C5H11SO7- | 11 | 0.11 | 5.47 | C6H4NSO6- | 1 | 0.01 | 0.48 |
| C9H9SO5- | 10 | 0.09 | 4.66 | Sum SCO | 207 | 2 | 100 |





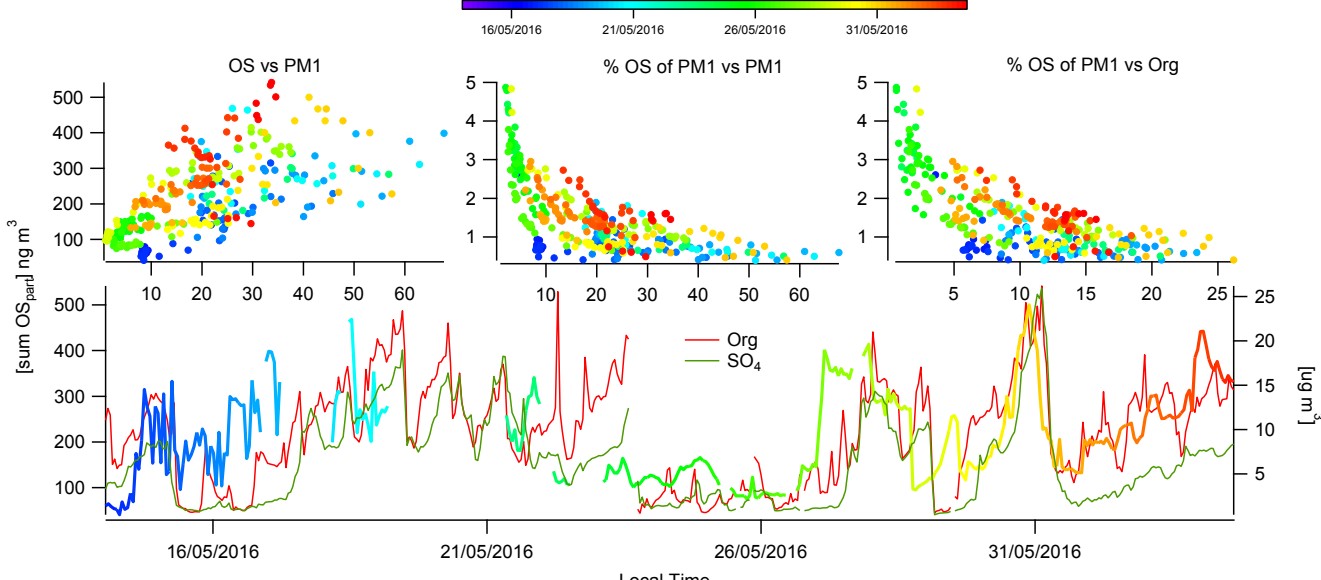

**Figure 3.** Time series of organics (green), SO$_4^{2-}$ (red) and total SCO (colour coded with time) displayed in the bottom panel. The correlation of SCO to PM$_1$, mass fraction of PM$_1$ and organics are displayed in the upper panel. The colour coding represents time throughout the campaign.





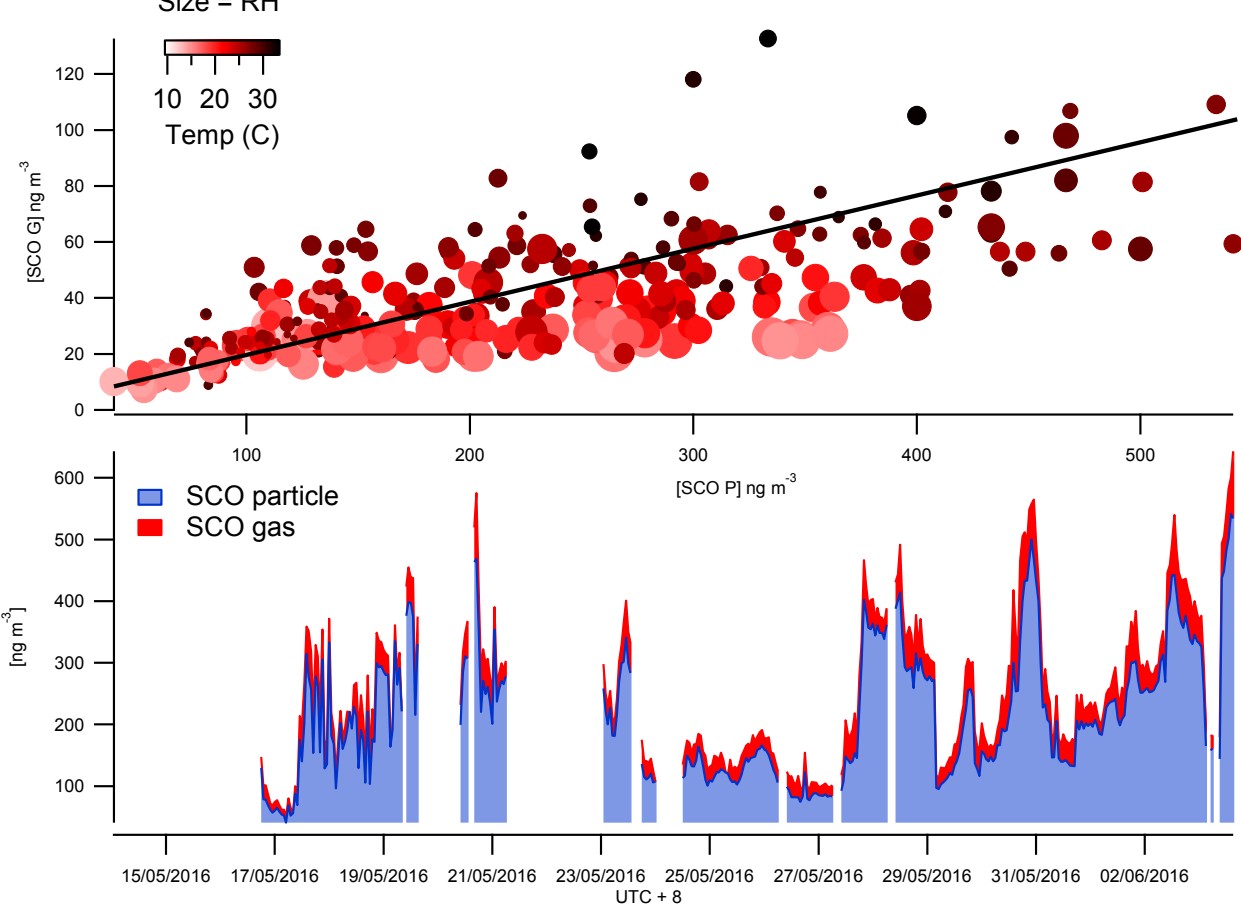

**Figure 4. Time series of total SCOs in the gas (SCOG) and particle (SCOP) phase (bottom panel) and their correlation colour coded by temperature and size binned by relative humidity (top panel).**



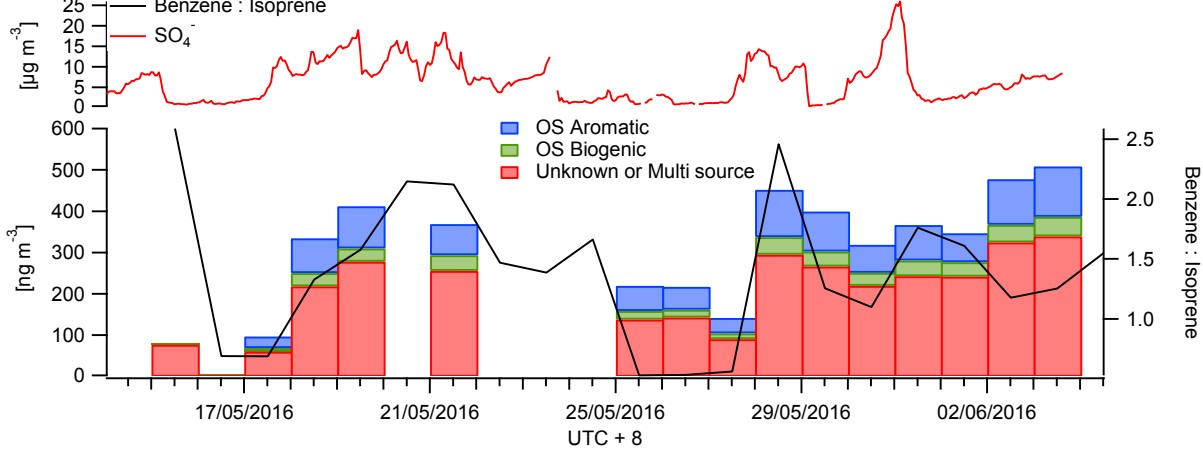

**Figure 5. A time series of the mean daily benzene to isoprene ratio as a marker for anthropogenic and biogenic influence (black) is displayed in the top panel. The CIMS data was also binned to provide mean daily SCO concentrations for aromatic (blue) and biogenic (green) precursor SCOs (bottom panel). The red bars represent SCOs with an unknown source or SCO produced via both biogenic and anthropogenic pathways. The AMS $SO_4^{2-}$ concentration is also presented to indicate availability of sulfur in the particle phase.**





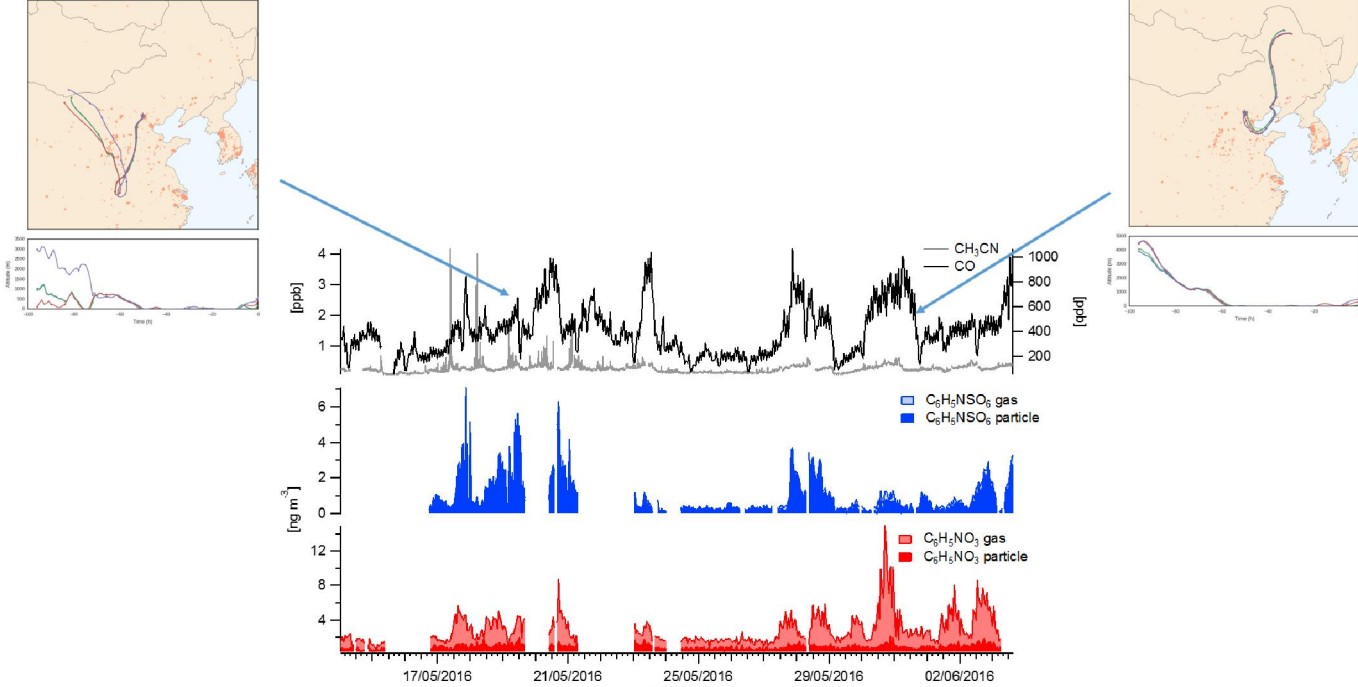

**Figure 6.** Time series of gas and particle phase NP (red), NP OS (blue) and gas phase acetonitrile (grey) and CO (black) between the 16th and 3rd June





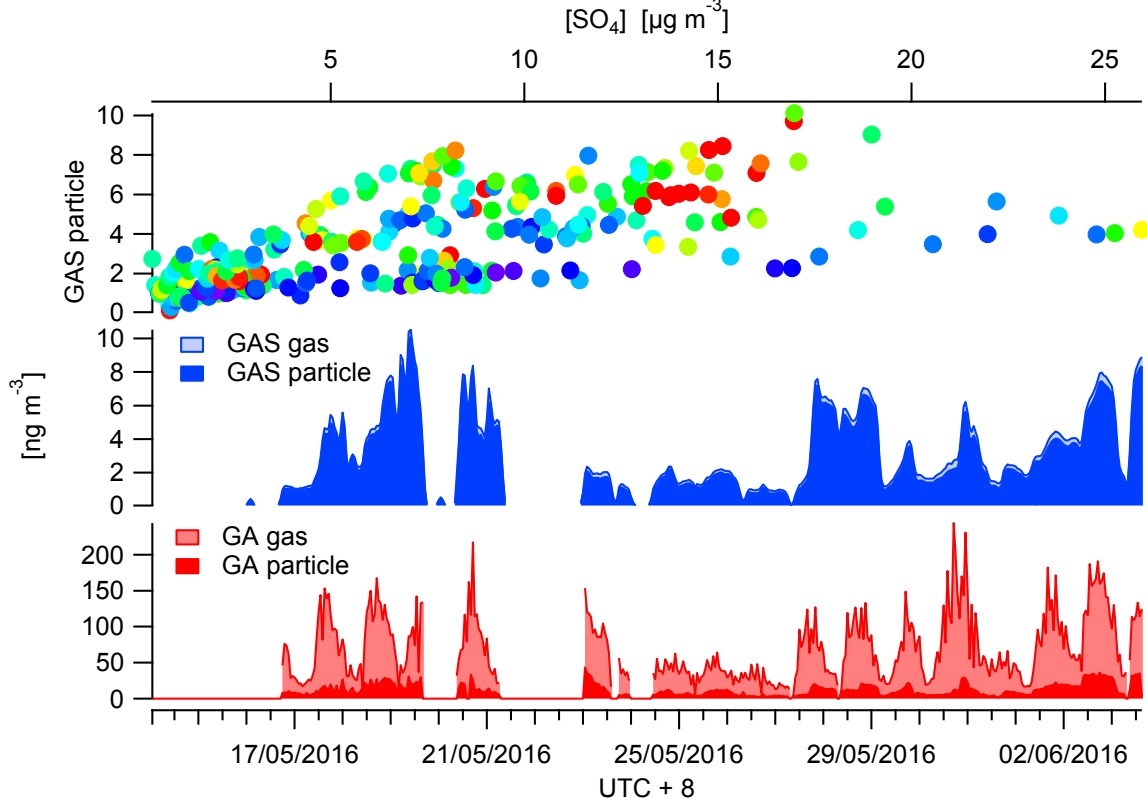

**Figure 7. Campaign time series of glycolic acid (red) and GAS (blue) in the particle and gas phase. The top panel illustrates the correlation between GAS in the particle phase and $SO_4^{2-}$ colour coded by GAp/GAg.**

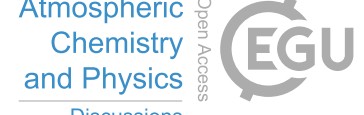



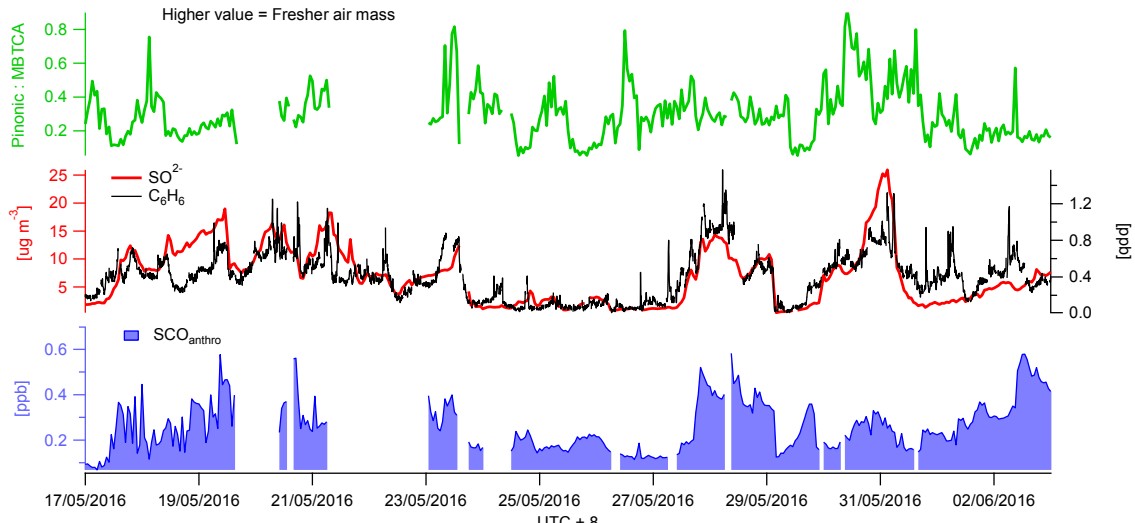

**(A)**

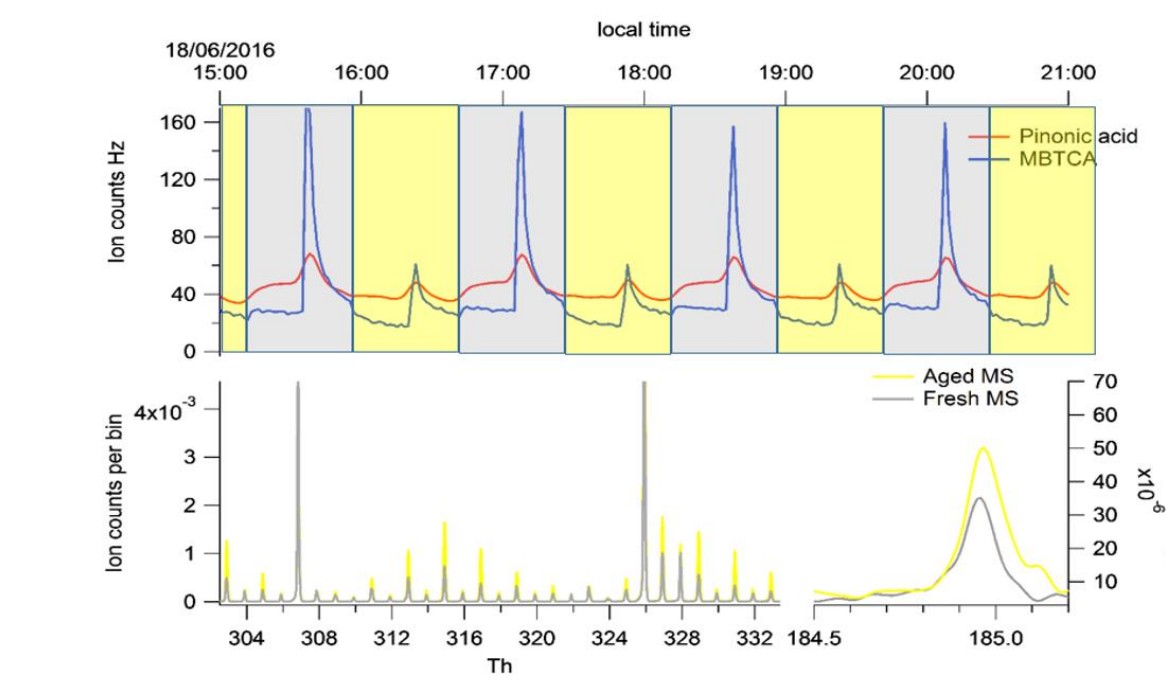

**(B)**

Figure 8. (A) Total benzene/PAH derived SCO (SCO$_{anthro}$) time series and respective SO$_4^{2-}$ and benzene concentrations. The indicator of photochemical aging (pinonic acid: MBTCA) is plotted in green. (B) illustrates the mass spectral difference between fresh and aged air masses through Go:PAM and respective time series for pinonic acid and MBTCA.





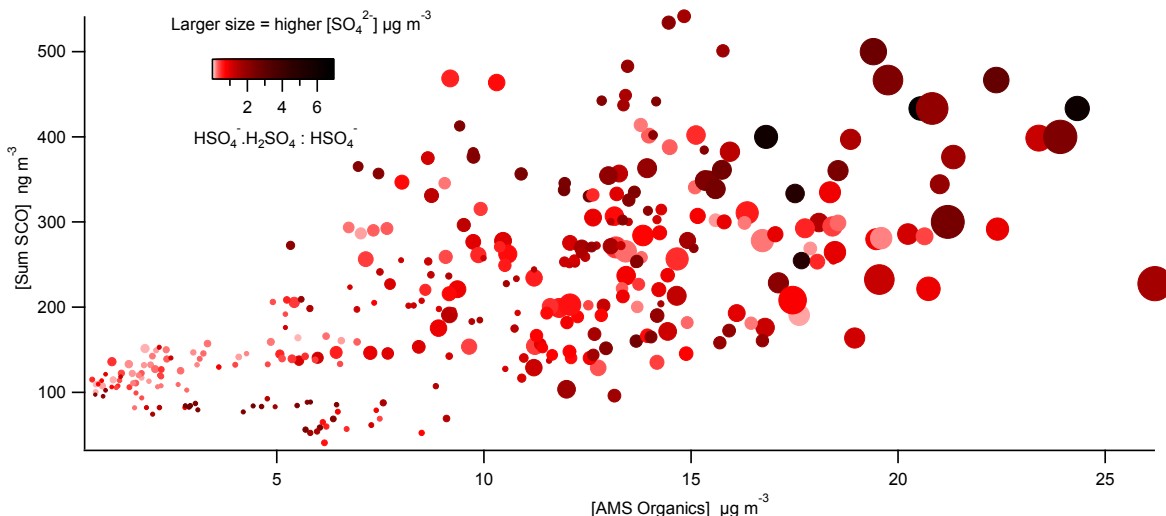

**Figure 9. Correlation plot of total SCOs vs total particle phase organics as a function of acidity (colour coding counts of $HSO_4^-.H_2SO_4 : HSO_4$) and $SO_4^{2-}$ (data point size spanning concentrations from 0.2 to 16 µg m$^{-3}$).**

