# Peer review of "Online gas and particle phase measurements of organosulfates, organosulfonates and nitrooxyorganosulfates in Beijing utilizing a FIGAERO ToF-CIMS"

_Atmospheric Chemistry and Physics, 2017_

## Referee Comment (RC1)

Online gas and particle measurements of organosulfates, organosulfonates and nitroxyorganosulfates in Beijing utilizing a FIGAERO ToF-CIMS

This paper describes the application of a FIGAERO ToF-CIMS to the characterisation of organic aerosol in Beijing, with a specific emphasis on the heteroatom containing CHOS and CHONS groups. The authors have attempted to quantify these species during a field campaign and then compare the temporal evolution to various chemical and metrological factors. I have no issue with the methods used to try and understand the data in the later part of the paper. While the idea has merit, and would be a very useful addition to the field, I cannot accept that the technique is actually measuring the species of interest based on the data provided in this paper. The extraction of very small and obscured signals from poorly resolved peaks, exemplified for the two OS species in Figure 1, has not been justified in any way. The description of the peak deconvolution is short and contains no evidence that this method has been validated. Have the authors measured the mass resolution to ensure that it really is 4000? The peaks widths used in the fitting require this to be known and the mass calibration across the entire range has to have sufficient accuracy. Has this approach been tested in the lab or are there previous publications? Also no uncertainties are provided. The entire paper and conclusions rests entirely on this component and as such I cannot recommend this paper be accepted to ACP at present. There is mention of a comparison to offline methods in the paper as being "outside the scope of this work". To me this is absolutely critical to provide validation of the method.

General comment:
Within the text both OS and SCO are used. Are these meant to be different things? It is hard to work out if they are being used interchangeably.
The results section contains a large number of typos and some very unclear sentences.

Specific comments

Abstract, line 33: "biogenic emissions contributed to only 19 % of the total SCO detected." While understand you want to make a split between these two sources, this is very much dependent of the spread of SCO you measure. Previous offline MS studies of OS in China, such as Wang et al., 2016 identified over 200 OS species in PM2.5. Therefore, your limited subset is very much biased depending on the choice of OS included, in this case only 17 species. You need to be very careful about making generalisation about the relative strength of the two sources based on this. Also, the C10H16NSO7 **ion** usually appears as a series of peaks in offline HPLC analysis and therefore is better described as monoterpene derived.

Page 6: SCO identification: There s not enough information here as outlined above. If this instrument has a mass resolution of 4000 (which is not explicitly stated) then at m/z 287, the minimum peak separation $\Delta M$, which allows two ion species to be distinguished, should be around 0.07. Thus in figure 1 (top, right), the light blue and yellow ions should be better resolved. How are the peak

centroids determined? The precision in which the intensity of very low s/n OS peaks (where the measured ion signal shows no evidence of this ion) can be retrieved is likely to be very poor. See Cubison and Jimenez, 2015.

Figure 1: The figure is difficult to understand and read. Why in the middle left hand panel have you not zoomed in so the labelled peaks can be observed? Also, the bottom plot does not really convey any information that is useful to the reader. The I- spectra seems irrelevant to the data being presented. Are there any peaks where the OS dominates the observed ion, rather than being a very small obscured peak?

Table 1 and 2: I am confused why there are two tables showing very similar information. Both tables contain a "mean" value but they are different? For example C11H11SO7 has the same mean %OA and %SCO in both tables but different mean concentrations (by a large amount 40 ng m$^{-3}$ v 120 μg m$^{-3}$)

Page 7, line 26: There doesn't appear to be any sulphur compounds in your reaction mixture?

Page 8, line 4: Figure 2 doesn't actually show a three point calibration. It shows the peaks obtained for three concentrations but it does show a calibration curve comparing concentration with response.

Page 8, line 14-21: I don't follow the reasoning that the low concentration of OS relative to the organic precursors results in little error. I would like to see some exampples of the double thermogram and know how widespread this effect is. Can you provide evidence that using only 1 species to determine the error is valid?

Page 8, line 27: This statement only holds true for species that desorb below 250 C.

Page 10, line 3: I do not understand this sentence at all. Quite often through the paper sentences are not very direct and contain many extra words.

Page 10, line 30: What does "mean presence" mean? Again this section lack clarity. I don't think a p:g ratio can be "prominent"? What is the 7.1 % referring to?

Page 11, section 4.1: I assume the PTR-MS measurements have been converted to daily averages? This is what the figure seems to present. The sentence starting on line 20 is very long and doesn't make sense. You are not measuring an attribution but using the measurements to test your attribution. Why do you give average toluene mixing ratios and then change to benzene? Be very clear here you are talking about your 17 SCO only.

Page 11, section 4.1.1: Green leaf volatiles and sesquiterpenes have also been identified as biogenic OS sources.

Page 12, section 4.1.2: I cannot see any of the trends you discuss here in Figure 6. You don't include any diurnal profiles, only a full time series and therefore the temporal evolution is not clear. At the end of the section I was confused as to whether you thought the NP OS concentration was driven by traffic (hence the second peak) or biomass burning? I guess in reality it's a combination of the two, but this needs to be clearer.

Figure 2: The egend says "time series" but none of the plots have a time axis? Should say these are m/z intensities. Is the average stick spectrum collected at the desorption temperature with the highest ion count?

Figure 3: The SCO times series coloured by time is really hard to see when sitting on top of the other signals. I would separate these out.

Figure 5: this legend needs work. The benzene to isoprene ratio is on the lower panel not the upper one. The AMS data is in the upper panel and should be stated. How does the anthropogenic SCO concentration change with the b:iso ratio? Most of the variability seems to be driven by the unknowns.

Refrence
Cubison, M. J. and Jimenez, J. L.: Statistical precision of the intensities retrieved from constrained fitting of overlapping peaks in high-resolution mass spectra, Atmos. Meas. Tech., 8, 2333-2345, https://doi.org/10.5194/amt-8-2333-2015, 2015.
Wang, X. K., Rossignol, S., Ma, Y., Yao, L., Wang, M. Y., Chen, J. M., George, C., and Wang, L.: Molecular characterization of atmospheric particulate organosulfates in three megacities at the middle and lower reaches of the Yangtze River, Atmos. Chem. Phys., 16, 2285-2298, https://doi.org/10.5194/acp-16-2285-2016, 2016

---

## Referee Comment (RC2) · Anonymous Referee #3 · 12 Dec 2017

This paper presents the characterization of organic aerosol sampled in Beijing using a FIGAERO ToF-CIMS with a focus on the organosulfates (CHOS and CHONS). In this manuscript, the authors have attempted to quantify these species and look at their distribution between gas and particle phases. While the method/idea proposed in this work is interesting and could lead to a new way to characterize such compounds, this paper cannot be accepted as it is. Indeed, the identification of the OSs is based on peak fittings that are highly questionable. In addition, the authors concluded on the validity of the method/results without any strong support/ evidence. As it is the paper

is speculative and major improvements are needed to support this work. The authors seem to have intentionally left crucial information out of this manuscript to write (an)other paper(s). If this current manuscript cannot stand by itself (and it is the case) additional information have to be added.

The first line of the abstract: "Fast" Inlet? The correct name of the FIGAERO is Filter inlet.

SCO identification/quantification: There is clearly a lack of information in these sections and more efforts are needed to better describe validate the method/identification:

- The mass calibration is crucial in the identification of the compounds, have the authors checked if the parameters didn't change over time? The authors should propose the time series of the parameters in the SI. - The authors need to provide the peak fittings for all of the peaks and not only two compounds. The peaks reported in Figure 1 are very small and poorly resolved, which make the "identification" questionable. If they claim they identified 17 SCOs, they should report 17 HR fittings. In addition, please add the masses and the formula of the compounds observed on the HR fittings. - It is hard to believe that the mass resolution of the instrument was 4000 based on the peak shape/HR fittings proposed in Figure 1. The authors should bring more evidence and make sure the resolution was the same throughout the campaign. - The authors mentioned that they have compared LC-MS and FIGAERO data. Where are these data? They need to be reported in this work. The authors said: "This analysis is not within the scope of this work and provides the basis of the correct identification to which a future paper will probe the caveats observed between the measurement techniques." It is actually the scope of the paper, demonstrating that the FIGAERO is able to quantify OSs. Therefore comparison with well-known techniques is more than crucial. LC-MS vs FIGAERO should be added ([C],...). - Page 8. Lines 9-12: The authors have to provide this information, that's an important parameter. In addition, we do not know if the ramp chooses during the field campaign leads to high fragmentation or not. -Page 8. Lines 13-15. The authors mentioned that the OS concentrations are relatively Interactive comment

low but in another section that they contribute to a significant fraction page 7 line 14. Make sure your statements are consistent. - Lines 14-19: The authors discussed the fragmentation issue. It is recognized that the FIGAERO or in general thermo desorption leads to the fragmentation of organic species (Thornton's group, Stark et al., 2017;...). While they acknowledged this problem, the authors didn't discuss this point when they determined/discussed the volatility of the OSs (Section 3.2). In the existing literature, previous FIGAERO studies have discussed this potential artifact and mention that the decomposition of oligomers could lead to lower Tmax than expected. Would it be the case for the OSs? The authors compare the volatility of acid compounds between the Knudsen and the FIGAERO. It is interesting but not relevant to the current study as they didn't quantify/look at the carboxylic acids. Have they done the comparison with the OSs used in this study (e.g.LAS & NP OS)? - The authors should provide the Tmax & thermograms for all the SCO and look at the evolution of individual Tmax throughout the campaign.

Page 9. The authors should provide simple analysis before going into too many details to validate the FIGAERO data (e.g. sum of organics vs OA; sum of organics vs SO4; sum of SCO vs OA, SO4,...).

Page 10. Lines 30-34. How is it possible that IEPOX-OS could be more volatile than less oxidized compounds, such as GAS? If it was the case previous measurements realized by Stone's group in the SE-US would have revealed such phenomena. Overall this discussion is lacking comparisons with previous studies (Lopez-Hilfiker et al., 2016; Hettiyadura et al., 2017) and evidence/better constraints to validate such "results".

Page 11. Lines 23-26: That is not true. Liu et al. 2017, ACP reported the formation of such OS from the photooxidation of cyclohexene.

---

## Author Response (AR1)

We thank all the reviewers for their valuable input and critical questions they have posed in which we feel we have answered fully and improved the manuscript in all areas questioned by the reviewers.

**Summary of amendments**

The main comments from both reviewers for this manuscript regarded the peak fitting and validating the Sulfur Containing Organic (SCO) measurement with the Iodide FIGAERO CIMS. We have addressed this by adding a section in the methodology that details the peak fitting, calibration and validation of the measurement via comparison with offline methods. We have described in further detail how the process of peak fitting with Tofware and how mass calibrants are utilised for accurate peak fitting across the full mass range, which is now illustrated in a supplementary figure displaying each mass calibrant ions error time series varying no greater than +/- 1 ppm for the whole campaign. Discussion of peak fitting the SCOs now details the "custom" peak shape which is utilised that does not assume a Gaussian peak shape in the spectra, further improving the ability to identify individual peaks under multi peak spectral fits and more accurately constrain the centroid value. The peak fitting is further discussed and peak fittings for all SCOs can be found in the supplementary data. The peak fittings here are average peak shapes from the entire field campaigns desorption data. Peak fitting can be performed in a number of different manners and in reality should be performed on many varying "periods" of data which fully represent the campaigns data. Here, we did not want to "cherry pick" the periods of higher SCO concentration to provide better peak fitting results, but rather highlight their significance in the spectra throughout the campaign, but also acknowledge the complexity of peak fitting and the consequential variation in possible error (or LOD issues) when fitting multi peaks. The validation of these peak fits were previously absent from the manuscript due to the comparison to offline methods being shown in an accompanying paper, although we have now included this analysis within the manuscript to illustrate how the peak fits presented can be utilised for accurate SCO measurements. A paragraph and plot have now been added to the manuscript detailing a time series comparison to offline HPLC and orbitrap measurements. The provided results show a good agreement between singular SCOs chosen, IEPOX sulfate and glycolic acid sulfate, with R values of 0.78 and 0.82 respectively. Comparison of the sum of SCOs measured by both orbitrap and HPLC also show good agreement (R = 0.7 and 0.81 respectively) further illustrating that the CIMS is agreeing with offline measurements of SCOs when the data is averaged to diurnal values. The method itself will contain a degree of error due to different locations on site (laboratory and inlet) and missing CIMS data in the analysis daily due to background measurements. The 2 calibrations are fully discussed that firstly prove the ionisation technique is linearly sensitive to SCOs, but also highlight that this instrument can quantify its measurements. Similarly for offline SCO measurement techniques, quantification is a limiting factor due to availability of standards, which also limits the ability for error quantification on the measurement. Here we utilise the two lab calibrants for quantification and error calculation, although we still do note that further work is necessary to improve future work on measurement of SCOs. It is worth noting that current CIMS literature utilises this approach due its ability to measure a vast range of species which do not have standards i.e. Lee *et al.,* (2016) who identify 87 organonitrates. We therefore feel this analysis now confirms to the reader that CIMS can measure SCOs and the subsequent analysis within the manuscript can be approached with confidence.

Fragmentation of SCOs is discussed in more detail citing the literature presenting possible fragmentation issues and subsequent complexities in VP calculations. We conclude that we see

fragmentation but stress we report fragmentation of SCOs into smaller ions and not oligomers fragmenting to SCOs of the same mass, which would be observed by a double Tmax desorption peak. We show a figure in the supplementary that illustrates the variation of Tmax for NP OS throughout the campaign and the average thermogram which exhibits no double desorption. We also state that fragmentation to the NP OS mass would require greater energy, which would increase the Tmax temperature and subsequent VP value. We acknowledge that this is the first field data of SCOs and interpretation of Tmax may be more problematic due to uncontrollable variations in field data. Therefore, we limit the analysis of Tmax and VPs to the calibrated NP OS, which we have performed under controlled conditions. The KEMS data supports the possibility for a gas phase presence of SCOs, which we indicate can be extracted from Tmax relative to that for NP OS, although we cannot explain the IEPOX sulfate high VP and therefore believe further analysis into this subject is wanted.

Detailed responses to each reviewer's comment are detailed below with corresponding actions taken within the manuscript.

**Anonymous Referee #3**

This paper presents the characterization of organic aerosol sampled in Beijing using a FIGAERO ToF-CIMS with a focus on the organosulfates (CHOS and CHONS). In this manuscript, the authors have attempted to quantify these species and look at their distribution between gas and particle phases. While the method/idea proposed in this work is interesting and could lead to a new way to characterize such compounds, this paper cannot be accepted as it is. Indeed, the identification of the OSs is based on peak fittings that are highly questionable. In addition, the authors concluded on the validity of the method/results without any strong support/ evidence. As it is the paper is speculative and major improvements are needed to support this work. The authors seem to have intentionally left crucial information out of this manuscript to write (an)other paper(s). If this current manuscript cannot stand by itself (and it is the case) additional information have to be added.

The first line of the abstract: "Fast" Inlet? The correct name of the FIGAERO is Filter inlet. SCO identification/quantification:

**Response: This is correct, it should read filter inlet**

**Action: This has been amended in the text**

There is clearly a lack of information in these sections and more efforts are needed to better describe validate the method/identification

- The mass calibration is crucial in the identification of the compounds, have the authors checked if the parameters didn't change over time? The authors should propose the time series of the parameters in the SI. - The authors need to provide the peak fittings for all of the peaks and not only two compounds. The peaks reported in Figure 1 are very small and poorly resolved, which make the "identification" questionable. If they claim they identified 17 SCOs, they should report 17 HR fittings. In addition, please add the masses and the formula of the compounds observed on the HR fittings. - It

is hard to believe that the mass resolution of the instrument was 4000 based on the peak shape/HR fittings proposed in Figure 1. The authors should bring more evidence and make sure the resolution was the same throughout the campaign. - The authors mentioned that they have compared LC-MS and FIGAERO data. Where are these data? They need to be reported in this work. The authors said: "This analysis is not within the scope of this work and provides the basis of the correct identification to which a future paper will probe the caveats observed between the measurement techniques." It is actually the scope of the paper, demonstrating that the FIGAERO is able to quantify OSs. Therefore comparison with well-known techniques is more than crucial. LC-MS vs FIGAERO should be added ([C],. . . ).

**Response: The above points are extremely important and have been accounted for in full within the manuscript to ensure the reader that SCOs are indeed being measured. We have now added a new section into the manuscript which aims to validate the measurements to the reader with the methods described by the reviewer. A time series mass calibration plot has been added to the supplementary and described in the text to show that the mass accuracy does not deviate greater or less than 1 and -1 throughout the campaign for the entire mass range. This provides the reader with confidence that correctly identified peaks should have little error related to mass accuracy and indeed peak identification is constantly as accurate as possible throughout the mass range. To aid this, the mass resolution of 3500 is quoted, which was indeed lower than optimum for the ToF CIMS, although was deemed necessary to increase the sensitivity as the X ray source provides a magnitude less ionisation ions than the usual Polonium 235 source.**

**Peak fittings of all OSs presented in the work have now been added to the supplementary text. The previous fits for figure 1 displayed the average peak fit for all the data over the whole campaign. To better represent the data we have now chosen the average peak fit for only the particle phase data. As we expect little OS in the gas phase, we would have a larger amount of gas phase peaks which would skew the significance of the SCOs. The SCO peaks now represent a significant peak on the spectrum highlighting the accurate peak fitting and representation in the particle phase on the spectra. We also note in the text that peak fitting is performed utilising a standard peak shape representing the data and not assuming a Gaussian peak. This better constrains the residual and the centroid of the peaks. We then calibrate for 2 SCOs to not only confirm the ability to ionise and detect SCOs but also show the ability of CIMS to quantify the measurement.**

**We have now also included the required comparison between the online and offline measurements of SCOs. A description of how the comparison is performed and example results for several OSs is displayed in supplementary plots. The SCOs measured by the CIMS and orbitrap correlated at the 0.05 significance level, which is stated in the text and again proves that both offline and online measurements significantly correlate. Correlations of example SCOs such as glycolic acid sulfate and IEPOX sulfate between HPLC and CIMS have an R of 0.78 and 0.82. The correlation between 5 overlapping SCOs results in an R value of 0.7.**

**We believe that by now presenting accurate peak fittings which do not deviate over time and a significant comparison with offline measurements of SCOS that the reader should be fully convinced that the ToF CIMS is indeed measuring SCOs and can therefore have confidence in the subsequent analysis and the ambient observations that is the focus of this manuscript.**

**Action: A new section has been added to manuscript providing ample evidence of accurate peak identification and quantification of SCOs. Firstly a mass calibrant time series has been provided in**

the supplementary showing the error varies less than +/-1 ppm throughout the campaign across the mass range. The mass resolution of 3500 is stated. All peak fittings are provided in the supplementary showing dominant SCO peaks in the spectrum. A comparison of the CIMS measurements to offline SCO measurements has been provided and detailed to show good agreement between the methods at a statistically significant level. The calibration of the 2 SCOs has been then placed into this section to provide further evidence for the reader that all caveats for identification and quantification have been accounted for.

Page 8. Lines 9-12: The authors have to provide this information, that's an important parameter. In addition, we do not know if the ramp chooses during the field campaign leads to high fragmentation or not.

**Response: The ramp rate was maintained in the field at the same rate as the calibration of the OS to ensure the same level of fragmentation. The varying degree of fragmentation was discovered post campaign during calibrations, therefore the choice of calibration applied was set to the same ramp rate as for the field measurements. The important message for the reader here is that changing ramp rate will change sensitivity due to fragmentation, which is important to account for during calibration.**

**Action: This has been clarified as explained in the response.**

Page 8. Lines 13-15. The authors mentioned that the OS concentrations are low but in another section that they contribute to a significant fraction page 7 line 14. Make sure your statements are consistent.

**Response: The contribution of OS to PM is relatively low but the representation of the OS measured here to total OS assumed from previous work is significant.**

**Action: This has been reworded to clarify its meaning.**

Lines 14-19: The authors discussed the fragmentation issue. It is recognized that the FIGAERO or in general thermo desorption leads to the fragmentation of organic species (Thornton's group, Stark et al., 2017;. . .). While they acknowledged this problem, the authors didn't discuss this point when they determined/discussed the volatility of the OSs (Section 3.2). In the existing literature, previous FIGAERO studies have discussed this potential artifact and mention that the decomposition of oligomers could lead to lower Tmax than expected. Would it be the case for the OSs? The authors compare the volatility of acid compounds between the Knudsen and the FIGAERO. It is interesting but not relevant to the current study as they didn't quantify/look at the carboxylic acids. Have they done the comparison with the OSs used in this study (e.g.LAS & NP OS)? - The authors should provide the Tmax & thermograms for all the SCO and look at the evolution of individual Tmax throughout the campaign.

**Response: The discussion of Tmax evolves from the observation of gas phase SCOs. The section discussing Tmax and VPs of the OSs has been reduced to solely discuss the OS calibrated by an independent instrument to confirm the possibility of gas phase OSs by having a semi volatile VP.**

The comparison to carboxylic acids serves solely to illustrate that compounds measured ambiently of known VP agree with the KEMS method and therefore provides reliability in scaling of the Tmax values to interpreted VPs. We now do not further probe VPs via Tmax analysis and state that this should firstly be performed in a laboratory, as was performed with NP OS as field data can provide further unresolvable complexities. Nevertheless the data still shows SCOs potential to be in the gas phase, supported by external VP measurements. If fragmentation was to occur by oligomers or dimers, the energy to break these bonds, then to vaporise the fragments would be greater than the energy needed to merely vaporise the OS mass detected and therefore would result in a second Tmax at higher temperatures than the one for OS we observe.

A figure illustrating the thermogram of NP OS and its Tmax values evolution throughout the campaign is now included as requested by the reviewer illustrating little variation in Tmax and no double desorption at this mass occurring from fragmentation.

Action: Further text has been provided to discuss fragmentation and the potential for it to affect determination of the VPs of SCOs, although we conclude that it would result in a higher Tmax for the SCOs, thus underestimating any estimation of VP. We also state no double desorption's are observed for the SCOs and a figure has been included to display stable Tmax values throughout the campaign.

Discussion of Tmax and VP is now limited to calibrated SOCs. The discussion on the comparison of carboxylic acid Tmaxs and VPs has been made more clear to represent them as a benchmark to validate the Tmaxs in this work illustrating the agreement with VPs from the KEMS, thus providing confidence in the KEMS SCO VP measurement.

Page 9. The authors should provide simple analysis before going into too many details to validate the FIGAERO data (e.g. sum of organics vs OA; sum of organics vs SO4; sum of SCO vs OA, SO4,. . .).

Response: The papers aim is to focus on organic sulphates sources and production in the ambient atmosphere. This is a novel approach and new analysis utilising this instrument. The CIMS is also specialised on single compound identification and not the sum of organics, so it likely to greatly underestimate the organic mass compared to OA. Therefore, we do not feel this analysis will provide further useful insights in the analysis of this work.

Action: No pre analysis has been placed into the paper due to the focus on SCOs. We have indicated in the paper that this analysis can be performed, although we are aware that the selectivity of the ionisation scheme inhibits CIMS to see all of the organic mass.

Page 10. Lines 30-34. How is it possible that IEPOX-OS could be more volatile than less oxidized compounds, such as GAS? If it was the case previous measurements realized by Stone's group in the SE-US would have revealed such phenomena. Overall this discussion is lacking comparisons with previous studies (Lopez-Hilfiker et al., 2016; Hettiyadura et al., 2017) and evidence/better constraints to validate such "results".

**Response: This is a very valid point to which we must address. Due to the limitation of laboratory calibrated SCOs (limited by standard availability), we have now limited the discussion of the VPs of SCOs. The calibration of NP OS via the KEMS instrument and FIGAERO indicate a semi volatile vapour pressure for the SCO. It is through extrapolation of this knowledge that other SCOs would have a relatively varying VP via comparison of their Tmax. Although we acknowledge that these other SCOs have not been calibrated, therefore we have removed the discussion of other SCOs VPs and table 3. We however believe the calibration using the KEMS instrument indicating the potential for gas phase SCOs is an important result from this work and can further lead to a new field of research both for VP calculations of SCOs utilising the FIGAERO and mechanistic understanding of SCO processes.**

**Action: The manuscript text is now limited to discussion of the VPs of SCOs which have been calibrated in the laboratory, NP OS. We then state that with proper calibration of SCOs, the Tmax and VPs can lead to further discussion of the utilisation of the FIGAERO for partitioning analysis of SCOs.**

Page 11. Lines 23-26: That is not true. Liu et al. 2017, ACP reported the formation of such OS from the photooxidation of cyclohexene.

**Response: This is correct and must be amended in the text**

**Action: The text has been amended to account for this**

**Anonymous Referee #1**

This paper describes the application of a FIGAERO ToF-CIMS to the characterisation of organic aerosol in Beijing, with a specific emphasis on the heteroatom containing CHOS and CHONS groups. The authors have attempted to quantify these species during a field campaign and then compare the temporal evolution to various chemical and metrological factors. I have no issue with the methods used to try and understand the data in the later part of the paper. While the idea has merit, and would be a very useful addition to the field, I cannot accept that the technique is actually measuring the species of interest based on the data provided in this paper. The extraction of very small and obscured signals from poorly resolved peaks, exemplified for the two OS species in Figure 1, has not been justified in any way. The description of the peak deconvolution is short and contains no evidence that this method has been validated. Have the authors measured the mass resolution to ensure that it really is 4000? The peaks widths used in the fitting require this to be known and the mass calibration across the entire range has to have sufficient accuracy. Has this approach been tested in the lab or are there previous publications? Also no uncertainties are provided. The entire paper and conclusions rests entirely on this component and as such I cannot recommend this paper be accepted to ACP at present. There is mention of a comparison to offline methods in the paper as being "outside the scope of this work". To me this is absolutely critical to provide validation of the method.

**Response: We have now added a new section into the manuscript which aims to validate the measurements to the reader with the methods described by the reviewer. A time series mass calibration plot has been added to the supplementary and described in the text to show that the**

mass accuracy does not deviate greater or less than 1 and -1 throughout the campaign for the entire mass range. This provides the reader with confidence that correctly identified peaks should have little error related to mass accuracy and indeed peak identification is constantly as accurate as possible throughout the mass range. To aid this, the mass resolution of 3500 is quoted, which was indeed lower than optimum for the ToF CIMS, although was deemed necessary to increase the sensitivity as the X ray source provides a magnitude less ionisation ions than the usual Polonium 235 source.

Peak fittings of all OSs presented in the work have now been added to the supplementary text. The previous fits for figure 1 displayed the average peak fit for all the data over the whole campaign. To better represent the data we have now chosen the average peak fit for only the particle phase data. As we expect little OS in the gas phase, we would have a larger amount of gas phase peaks which would skew the significance of the SCOs. The SCO peaks now represent a significant peak on the spectrum highlighting the accurate peak fitting and representation in the particle phase on the spectra.

We have now also included a necessary comparison between the online and offline measurements of SCOs. A description of how the comparison is performed and example results for several OSs is displayed in supplementary plots. The SCOs measured by the CIMS and orbitrap correlated at the 0.05 significance level, which is stated in the text and again proves that both offline and online measurements significantly correlate.

Calibration of the 2 SCOs then follows the comparison to further illustrate the ability of the CIMS to accurately measure and quantify the SCOs. The laboratory calibration of NP OS in the paper highlight the ability for CIMS to measure OSs and with a high sensitivity, which would have also previously been assumed as the iodide ionisation scheme has a high affinity to sulphuric acid.

The uncertainty for the calibrated NP OS is stated to apply for all OS. We acknowledge in the paper that without direct calibration it is difficult to provide an accurate error for all species. This is similar to that applied for 87 organonitrates by Lee et al. (2016) therefore we feel that this method is quite standard for this instrument within the literature.

We believe that by now presenting accurate peak fittings which do not deviate over time and a significant comparison with offfline measurements of SCOS that the reader should be fully convinced that the ToF CIMS is indeed measuring SCOs and can therefore have confidence in the subsequent analysis.

Action: This is the same addition to the responses to reviewer 1:

A new section has been added to manuscript providing ample evidence of accurate peak identification and quantification of SCOs. Firstly a mass calibrant time series has bene provided in the supplementary showing the error varies less than +/-1 ppm throughout the campaign across the mass range. The mass resolution of 3500 is stated. All peak fittings are provided in the supplementary showing dominant SCO peaks in the spectrum. A comparison of the CIMS measurements to offline SCO measurements has bene provided and detailed to show good agreement between the methods at a statistically significant level. The calibration of the 2 SCOs has been then placed into this section to provide further evidence for the reader that all caveats for identification and quantification have been accounted for.

General comment:

Within the text both OS and SCO are used. Are these meant to be different things? It is hard to work out if they are being used interchangeably. The results section contains a large number of typos and some very unclear sentences.

**Response: This has been acknowledged and will be amended in the text**

**Action: The use of SCO and OS has been adjusted in the text to better represent its use i.e. to define possible structural differences in the correct context**

**Specific comments**

Abstract, line 33: "biogenic emissions contributed to only 19 % of the total SCO detected." While understand you want to make a split between these two sources, this is very much dependent of the spread of SCO you measure. Previous offline MS studies of OS in China, such as Wang et al., 2016 identified over 200 OS species in PM2.5. Therefore, your limited subset is very much biased depending on the choice of OS included, in this case only 17 species. You need to be very careful about making generalisation about the relative strength of the two sources based on this. Also, the $C_{10}H_{16}NSO_7$ ion usually appears as a series of peaks in offline HPLC analysis and therefore is better described as monoterpene derived.

**Response: This is a very important factor which we must address better. The offline measurements of OSs by orbitrap and HPLC come to a similar conclusion, which includes a large suite of OSs detected. The parallel offline analysis utilises 34 OSs for their analysis and found that anthropogenic OSs were 4 times more abundant than the biogenic OSs.**

**Action: The text now acknowledges this potential issue but provides information that the offline methods also come to this conclusion upon analysis of the 34 OSs utilised in analysis.**

Page 6: SCO identification: There's not enough information here as outlined above. If this instrument has a mass resolution of 4000 (which is not explicitly stated) then at m/z 287, the minimum peak separation ΔM, which allows two ion species to be distinguished, should be around 0.07. Thus in figure 1 (top, right), the light blue and yellow ions should be better resolved. How are the peak centroids determined? The precision in which the intensity of very low s/n OS peaks (where the measured ion signal shows no evidence of this ion) can be retrieved is likely to be very poor. See Cubison and Jimenez, 2015.

**Response: The average mass resolution for a standard peak throughout the campaign was 3500, which was lower than optimum (4000) due to a sacrifice of resolution to increases sensitivity for tuning with the new X ray ionisation source, which provides a magnitude less ionisations ions than Polonium210. The peak fittings in the top panel of figure were single point peak fits for the whole dataset. These have now been replaced with more representative standard peak fittings for the data, which are for particle phase measurements only (not including gas phase peak fittings as before which would always include much lower OS counts).**

**Action: The mass resolution of 3500 is stated in the text and description of the peak fitting and what spectra are used is more accurately described. We state the current issue with CIMS that during analysis peak fittings and accuracy may change due to the changing intensity of co-existing peaks under the same bulk peak. We do however fit a custom peak shape to the spectra to ensure that the residual is not assuming a Gaussian fit, creating a higher accuracy when fitting a centroid to the peak position.**

Figure 1: The figure is difficult to understand and read. Why in the middle left hand panel have you not zoomed in so the labelled peaks can be observed? Also, the bottom plot does not really convey any information that is useful to the reader. The I- spectra seems irrelevant to the data being presented. Are there any peaks where the OS dominates the observed ion, rather than being a very small obscured peak? Table 1 and 2: I am confused why there are two tables showing very similar information. Both tables contain a "mean" value but they are different? For example $C_{11}H_{11}SO_7$ has the same mean %OA and %SCO in both tables but different mean concentrations (by a large amount 40 ng m-3 v 120 μg m-3)

**Response: We agree with this comment and will amend the tables and figure to represent the data in a more concise, clear and informative manner**

**Action: Table 1 and 2 have now been merged and the data represented should be more clear and concise. Figure 1 has now been amended to remove the whole I- spectrum. The peaks displayed in the top panels now illustrate peaks from high and low masses on the spectrum. The peak fittings now represented in the supplementary data show SCO ions that have significant peak counts as we now exclude gas phase spectra from this analysis.**

Page 7, line 26: There doesn't appear to be any sulphur compounds in your reaction mixture?

**Response: The mixture stated as trioxide pyridine was actually sulphur trioxide pyridine.**

**Action: This information has been added to the text**

Page 8, line 4: Figure 2 doesn't actually show a three-point calibration. It shows the peaks obtained for three concentrations but it does show a calibration curve comparing concentration with response.

**Response: A 3 point calibration is now shown in the figure**

**Action: Figure 2 has been amended to now illustrate the thermogram for 3 different mass loadings on the filter and their respective spectra highlighting peaks observed during desorption and also a 4 point calibration curve (1 extra point for background desorption) with NP OS molecules plotted on the x axis and ion counts per desorption on the y axis.**

Page 8, line 14-21: I don't follow the reasoning that the low concentration of OS relative to the organic precursors results in little error. I would like to see some examples of the double thermogram and know how widespread this effect is. Can you provide evidence that using only 1 species to determine the error is valid?

**Response: This was caused due to typographic error within the text. The text was supposed to state little contribution of organic peak produced relative the OS (i.e. it was reported the wrong way around)**

**Error calculations are difficult without laboratory standards for calibrations to accurately calculate the error. It is common practise within CIMS literature to utilise a calibration for a functional group (such as organonitrates in Lee et al., 2016) and with that assume the same error. We therefore have applied a large error to account for this possibly varying parameter.**

**Action: This text has been amended and the application of one error constant has been explained more concisely.**

Page 8, line 27: This statement only holds true for species that desorb below 250 C

**Response: This point is valid and must be stated within the text.**

**Action: This has now been acknowledged within the text.**

Page 10, line 3: I do not understand this sentence at all. Quite often through the paper sentences are not very direct and contain many extra words.

**Response and action: This sentence has been removed due to extensive changed within this section prompted by the reviewer's comments regarding potential inaccuracies of gas phase SCO presence. Overall, the text has been revised to reduce the numbers of words and make the text more concise and effective**

Page 10, line 30: What does "mean presence" mean? Again this section lack clarity. I don't think a p:g ratio can be "prominent"? What is the 7.1 % referring to?

**Response and action: This line has now been removed due to the removal of VP analysis of uncalibrated SCOs**

Page 11, section 4.1: I assume the PTR-MS measurements have been converted to daily averages? This is what the figure seems to present. The sentence starting on line 20 is very long and doesn't make sense. You are not measuring an attribution but using the measurements to test your attribution. Why do you give average toluene mixing ratios and then change to benzene? Be very clear here you are talking about your 17 SCO only.

**Response: Yes they have been converted to daily averages. The inclusion of toluene was a typographic error.**

**Action: This sentence has been clarified and now correctly reports PTR-MS measurements of benzene and isoprene. It now reads "Thus, as shown in Figure 5, PTR-MS measurements mean daily concentrations of benzene and isoprene were utilized to evaluate if the ratio benzene and isoprene can be related to the contribution of aromatic and biogenic SCOs measured in this work"**

Page 11, section 4.1.1: Green leaf volatiles and sesquiterpenes have also been identified as biogenic OS sources.

**Response and action: This work has now been acknowledged and the reference to Shalmzari et al., (2014) has been added to the text.**

Page 12, section 4.1.2: I cannot see any of the trends you discuss here in Figure 6. You don't include any diurnal profiles, only a full time series and therefore the temporal evolution is not clear. At the end of the section I was confused as to whether you thought the NP OS concentration was driven by traffic (hence the second peak) or biomass burning? I guess in reality it's a combination of the two, but this needs to be clearer.

**Response: The NP OS from biomass burning from more aged air masses has had time to oxidise and produce the NP OS whereas local sources of NP (biomass burning and traffic) indeed can be sources of NP OS but as they are fresher sources, are less oxidised, and therefore produce much lower signals of NP OS compared to the aged BB air masses.**

**Action: NP AND NP OS diurnal profiles have now been added to the supplementary data to support the analysis within this section. The text has been made clearer to indeed indicate that both traffic and biomass burning can be a source of SCOs, although the local sources are fresher and hence provide lower mixing ratios of the secondary product.**

Figure 2: The legend says "time series" but none of the plots have a time axis? Should say these are m/z intensities. Is the average stick spectrum collected at the desorption temperature with the highest ion count?

**Response and action: The caption now reads" Figure 2. The desorption profile of NP OS 3 step calibrations for 0.1 μl, 0.2 μl and 0.3 μl 1000 ppm solution is displayed in the bottom panel and its corresponding average stick spectrum (top left) and sum of counts per molecule loading for each calibration (top right)". The spectrum is an average of the whole desorption profile.**

Figure 3: The SCO times series coloured by time is really hard to see when sitting on top of the other signals. I would separate these out.

**Response and action: Figure 3 has been amended to display the SCOs on a separate plot panel to the AMS organic and sulfate data.**

Figure 5: this legend needs work. The benzene to isoprene ratio is on the lower panel not the upper one. The AMS data is in the upper panel and should be stated. How does the anthropogenic SCO concentration change with the b:iso ratio? Most of the variability seems to be driven by the unknowns.

**Response and action: Figure 5 has been amended so the legends are now clear and concise. Although not a dominating factor in the plot, the anthropogenic SCOs also follow a similar time series trend as the benzene:isoprene ratio, similar to what can be seen by the unknowns.**

[revised manuscript text omitted]

---

## Author Response (AR2)

**Response to editor's revisions on "Chlorine oxidation of VOCs at a semi-rural site in Beijing: Significant chlorine liberation from ClNO2 and subsequent gas and particle phase Cl-VOC production" by Michael Le Breton et al.**

Main text:

Page 1, lines 5 and 6: Should the full stop (".") after "Chak" and "Carl" not be removed? Error removed

Page 1, line 6: The comma after "Zhu9" should not be in superscript. Error removed

Page 1, line 17: Replace "(School" by "School". Error corrected

Page 1, line 25: Replace "were" by "was". Error corrected

Page 1, line 26: Abbreviations and acronyms, here "HPLC" should be defined (written full-out) when first used. Now abbreviated

Page 1, line 27: Replace "maxima" by "maximum". Replaced

Page 1, line 30: Abbreviations and acronyms, here "RH" should be defined (written full-out) when first used. Replaced

Page 1, line 32: Replace "analysed" by "measured". Replaced

Page 1, line 33: Replace "representing" by "represented" and replace "Anthropogenic" by "anthropogenic". Replaced

Page 1, line 34: Abbreviations and acronyms, here "PAH" should be defined (written full-out) when first used. Now abbreviated

Page 2, line 4: Replace "significant" by "a significant". Replaced

Page 2, line 15: "Pope et al. (2011)" is missing in the Reference list. Pope 2011 removed from text

Page 2, line 17: Replace "remains" by "remain". Replaced

Page 2, line 18: Replace "its chemical" by "their chemical". Replaced

Page 3, line 2: Replace "e.g." by "e.g.,". Replaced

Page 3, line 13: Replace "an acidic" by "acidic". Replaced

Page 3, line 28: Replace "filters taken" by "filters". Replaced

Page 3, line 31: Replace "has also" by "have also". Replaced

Page 4, line 4: Replace "et al., (" by "et al. (". Replaced

Page 4, line 14: Replace "and University" by "and the University". Replaced

Page 4, line 19: Replace "and humidity" by "and relative humidity". Replaced

Page 4, lines 20-22: It should be indicated for which size fraction this sentence applies. For PM1

Page 4, line 22: Replace "contained high" by "contained a high". Replaced

Page 4, line 27: Replace "were mostly" by "were mostly with". Replaced

Page 5, line 2: Replace "et al., (" by "et al. (". Replaced

Page 5, line 8: Abbreviations and acronyms, here "UHP" should be defined (written full-out) when first used. Now abbreviated

Page 5, line 10: Abbreviations and acronyms, here "IMR" should be defined (written full-out) when first used. Now abbreviated

Page 7, line 1: Abbreviations and acronyms, here "NOS" should be defined (written full-out) when first used. Now abbreviated

Page 7, line 1: Replace "le Breton" by "Le Breton". Replaced

Page 7, line 8: Replace "NOS's" by "NOSs". Replaced

Page 7, line 10: Replace "as Olson" by "as used by Olson". Replaced

Page 7, line 11: Replace "(0.96g, 7.75mmol) in 2mL" by "(0.96 g, 7.75 mmol) in 2 mL". Replaced

Page 7, line 18: Abbreviations and acronyms, here "HR" should be defined (written full-out) when first used. Now abbreviated

Page 7, line 21: Replace "were calculated" by "was calculated". Replaced

Page 7, line 24: Replace "i.e." by "i.e.,". Replaced

Page 7, line 25: Replace "were performed" by "was performed". Replaced

Page 7, line 26: Replace "highlighted an" by "it was found that an". Replaced

Page 7, line 33: Replace "desorption's" by "desorptions". Replaced

Page 8, line 2: Replace "e.g. Lee et al., (" by "e.g., Lee et al. (". Replaced

Page 8, line 3: Abbreviations and acronyms, here "ON" should be defined (written full-out) when first used. Now abbreviated

Page 8, line 12: Replace "e.g." by "e.g.,". Replaced

Page 8, line 16: Replace "over corresponding" by "over the corresponding". Replaced

Page 8, line 32: Replace "Pascals" by "Pascal". Replaced

Page 9, line 2: Abbreviations and acronyms, here "VP" should be defined (written full-out) when first used. Now abbreviated

Page 9, line 9: Replace "Pa. (Booth" by "Pa (Booth". Replaced

Page 9, line 21: Abbreviations and acronyms, here "OA" should be defined (written full-out) when first used in the body of the manuscript. Now abbreviated

Page 9, line 22: Replace "that SCO" by "that the SCO". Replaced

Page 9, line 25: Replace "that SCO" by "that the SCO". Replaced

Page 10, line 5: Replace "than absolute" by "than the absolute". Replaced

Page 10, line 10: Replace "e.g." by ", e.g.," and replace "(e.g." by "(e.g.,". Replaced

Page 10, line 14: Replace "i.e.," by ", i.e.,". Replaced

Page 10, line 19: Replace "e.g." by "e.g.,". Replaced

Page 10, line 27: Replace "(Pa)" by "Pa". Replaced

Page 10, lines 31-32: "Bilde et al. (2015)" is missing in the Reference list. Added to reference list

Page 10, line 32: Replace ". Using" by " Pa. Using". Replaced

Page 10, line 33: Replace "were it" by "where it". Replaced

Page 11, line 2: Replace "need to" by "needs to". Replaced

Page 11, line 5: "Lopez-Hilfiker et al. (2016)" is missing in the Reference list. Added to reference list

Page 11, line 9: Insert a space before "(SI4)" and replace "14 Celisus" by "14 degrees Celsius". Replaced

Page 11, line 10: Replace "et al., (" by "et al. (". Furthermore, "Huang et al. (2017)" is missing in the Reference list. Replaced and added

Page 11, line 11: Replace "assess" by "assess the". Replaced

Page 11, line 14: Replace "as temperature" by "as the temperature". Replaced

Page 11, line 25: Replace "some which have" by "some have" and replace "e.g." by "e.g.,". Replaced

Page 12, line 1: Replace "ratio benzene" by "ratio between benzene". Replaced

Page 12, line 11: Place the second "et al." in italic and place "2015" not in italic. Replaced

Page 12, line 21: Delete either "polyaromatic hydrocarbon" or "(PAH)". As indicated above the acronym "PAH" should be defined (written full-out) when first used. Deleted

Page 13, line 6: Replace "6am" by "6 am". Replaced

Page 13, line 10: Replace "4pm" by "4 pm". Replaced

Page 13, line 12: Replace "4pm" by "4 pm". Replaced

Page 13, line 16: Replace "8am" by "8 am". Replaced

Page 13, line 32: Abbreviations and acronyms, here "GA" should be defined (written full-out) when first used. Now abbreviated

Page 14, line 2: Replace "concentrations" by "concentration", replace "high fraction" by "a high fraction" and replace "in particle" by "in the particle". Replaced

Page 14, line 3: Replace "provides higher" by "provides a higher". Replaced

Page 14, line 5: "Liao et al. (2016)" is missing in the Reference list.

Page 14, line 7: Replace "between GAp" by "is observed between GAp". Replaced

Page 14, line 12: Replace "i.e." by "i.e.,". Replaced

Page 14, line 20: Replace "and Amazon" by "and the Amazon". Replaced

Page 15, line 26: Replace "regards to" by "regard to". Replaced

Page 16, line 6: Replace "to a small" by "a small" and replace "was dominated" by "were dominated". Replaced

Page 16, line 8: Replace "abundant SCO" by "abundant SCOs". Replaced

Page 16, line 23: Replace "by Swedish" by "by the Swedish". Replaced

Page 17, lines 12-14: There is not referred to this reference within the text. Deleted

Page 17, line 18: Replace "Claevs" by "Claeys". Replaced

Page 17, lines 29-34: There is not referred to this reference within the text. Deleted

Page 20, lines 1-3: There is not referred to this reference within the text. Deleted

Page 20, lines 15-20: This reference should be moved down to before "Li et al., 2017". Moved

Page 22, lines 4-6: "Shalamzari et al. (2013)" should be removed as it is already given in lines 1-3. Deleted

Page 22, line 23: Replace "2017" by "2007". Replaced

Page 22, lines 30-32: There is not referred to this reference within the text. Deleted

Page 23, lines 12: This reference should be moved down to after "Zhang et al., 2012a". Moved
Page 25, within Table 1: Replace "Surrat" by "Surratt" on 5 occasions. Replaced
Page 26, line 10: Delete "dat" or move it down and replace it by "date". Deleted
Page 26, line 12: Replace "Time series" by "The time series". Replaced

Supplement:
Caption of Figure SI4, second line: Replace "top left window" by "top right window". Replaced

---

## Author Response (AR3)

**Response to editor's revisions on "Chlorine oxidation of VOCs at a semi-rural site in Beijing: Significant chlorine liberation from ClNO2 and subsequent gas and particle phase Cl-VOC production" by Michael Le Breton et al.**

All the below editor's comments have been amended appropriately within the text

Comments to the Author:

The comments given below should be addressed and the following alterations are needed for the Main text before the manuscript can be published in ACP:

Page 1, line 26: Replace "Chromotography" by "Chromatography".

Page 1, line 33: Replace "Anthropogenic" by "anthropogenic".

Page 2, line 31, and on 18 other occasions further in the manuscript: Replace "Surrattt" by "Surratt".

Page 4, line 25: Replace "contained high" by "contained a high".

Page 7, line 11: Replace "as Olson" by "as used by Olson".

Page 7, line 13: Replace "in 2mL" by "in 2 mL".

Page 10, line 12: Replace "of e.g., gas" by "of, e.g., gas".

Page 11, line 11: Replace "Celisus" by "Celsius".

Page 17, lines 18-20: There is not referred to this reference within the text.

Page 18, lines 1-5: There is not referred to this reference within the text.

Page 20, lines 4-6: There is not referred to this reference within the text.

Page 21, lines 14-15: The title of the journal article should be in lower case instead of in Title Case.

Page 22, line 11: The reference to "Shalamzari et al. (2014)" should start on a new line.

Page 23, lines 1-3: There is not referred to this reference within the text.

Page 23, line 11: This reference should be moved down to after "Zhang, Q. et al., 2007".

Page 25, within Table 1: Replace "Surrat" by "Surratt" on 5 occasions.

---

## Author Response (AR4)

**Response to editor's revisions on "Chlorine oxidation of VOCs at a semi-rural site in Beijing: Significant chlorine liberation from ClNO2 and subsequent gas and particle phase Cl-VOC production" by Michael Le Breton et al.**

Page 4, line 25: Replace "contained agh" by "contained a high".

The text has been amended